# Histology and Ultrastructure of the Nephron and Kidney Interstitial Cells in the Atlantic Salmon (*Salmo salar* Linnaeus 1758) at Different Stages of Life Cycle

**DOI:** 10.3390/biology12050750

**Published:** 2023-05-19

**Authors:** Ekaterina A. Flerova, Victoria V. Yurchenko, Alexey A. Morozov, Evgeniy G. Evdokimov, Alena A. Bogdanova, Maksim Yu. Alekseev, Dmitry S. Sendek, Sergey F. Titov

**Affiliations:** 1Faculty of Biology and Ecology, P.G. Demidov Yaroslavl State University, Sovetskaya St. 14, Yaroslavl 150003, Russia; skrad200052@yandex.ru; 2Papanin Institute for Biology of Inland Waters, Russian Academy of Sciences, IBIW 109, Borok 152742, Russia; victoria.yurchenko@rambler.ru (V.V.Y.); aleksey.a.morozov@gmail.com (A.A.M.); 3Yaroslavl Scientific Research Institute of Livestock Breeding and Forage Production—Federal State Budget Sciences Institution “Federal Williams Research Center of Forage Production and Agroecology”, Lenin St. 1, Mikhailovskiy, Yaroslavl 157517, Russia; bogdanova.ale@gmail.com; 4Polar Branch of FSBSI “VNIRO” (“PINRO” Named after N.M. Knipovich), 6 Akademika Knipovicha St., Murmansk 183038, Russia; m.alexeev09@mail.ru; 5Federal Agency for Fishery, Russian Federal “Research Institute of Fisheries and Oceanography” “VNIRO”, Saint Petersburg Branch of VNIRO (“GosNIORKH” Named after L.S. Berg), Naberezhnaya Makarova St. 26, St. Petersburg 199053, Russia; sendek@mail.ru (D.S.S.); sergtitov_54@mail.ru (S.F.T.)

**Keywords:** anadromous fish, salmonidae, migration, renal corpuscle, renal tubule, ionocytes, immune cells, mesonephros

## Abstract

**Simple Summary:**

This article presents new data on the histology and ultrastructure of the trunk kidney in two isolated populations of Atlantic salmon, from the Baltic and Barents Seas, covering both the freshwater and the seawater period of their life cycle. We studied juvenile salmon living in freshwater (parr), juvenile salmon running into the sea (smolts), adults feeding in the sea, adults returning to their natal river to spawn, and spawning individuals. According to the results obtained, the ultrastructural changes in the nephron, believed to affect the glomerular filtration rate and the amount of urine produced in salmon kidneys, occurred as early as the smolting stage. Such alterations reflect fundamental changes in the course of pre-adaptation to life in salt water. In the Barents Sea population, the adult salmon caught in the sea had the most pronounced ultrastructural changes, which are characteristic of marine fish. In the salmon that entered the river mouth and stayed in the fresh water for less than a day, structural rearrangements occurred only in the distal tubules of the nephron. The salmon from the Baltic Sea population had less pronounced differences in the morphometric characteristics of the nephron among the studied ontogenetic stages, apparently because of the low salinity in the sampling area. In both studied populations, the activation of cellular immunity was initiated during the parr–smolt transformation. Another distinct immune response was registered in the adults returning to the river for spawning.

**Abstract:**

This article presents data on the mesonephros histology and ultrastructure in the Atlantic salmon from the Baltic Sea and Barents Sea populations, with an emphasis on comparisons between the following ontogenetic stages: parr, smolting, adult life at sea, the adults’ return to their natal river to spawn, and spawning. The ultrastructural changes in the renal corpuscle and cells of the proximal tubules of the nephron occurred as early as the smolting stage. Such changes reflect fundamental alterations during the pre-adaptation to life in saltwater. In the Barents Sea population, the adult salmon sampled in the sea had the smallest diameters of the renal corpuscle and proximal and distal tubules, the most narrow urinary space, and the thickest basement membrane. In the group of salmon that entered the mouth of the river and spent less than 24 h in freshwater, the structural rearrangements occurred only in the distal tubules. Better development of the smooth endoplasmic reticulum and a greater abundance of mitochondria in the tubule cells were observed in the adult salmon from the Barents Sea compared to those from the Baltic Sea. Cell-immunity activation was initiated during the parr–smolt transformation. Another pronounced innate-immunity response was registered in the adults returning to the river to spawn.

## 1. Introduction

As migratory fish, Atlantic salmon spend their juvenile phase in freshwater habitats. In rivers, subadults grow and develop, preparing for migration into the sea, where they spend one to several years. The marine habitat provides adult fish with ample feeding resources, ensuring energy for somatic growth and gonadal development for their return to their river of origin to spawn [1]. To pass through habitat transitions, anadromous salmon must adapt to many external factors that lead to changes in certain physiological traits, beginning during their migration toward the sea and coming to an end when the fish migrate back to the river [2,3]. Migrating salmon exhibit high adaptive capacity during these habitat shifts [3,4,5,6,7]. The vast majority of studies on salmon migration are related to physiological changes in the osmoregulatory system, with only a few studies on the ultrastructural changes in the nephron [4,8,9,10].

As a unique hematopoietic organ in which cells proliferate and mature, and in which all the stages of the immune response can be completed independently of other lymphoid organs, the kidneys play a crucial role in osmoregulation and make a significant contribution to the nonspecific defense system. Therefore, studies of its structure are important from a comparative evolutionary perspective [11,12,13]. Previously, changes in the essential kidney excretory units, depending on sex and season, were registered in adults of *Salmo trutta* f. fario Linnaeus 1758 at a light-microscopic level using stereological methods [13]. Furthermore, some studies employed transmission-electron microscopy to describe the histology and ultrastructure of the trunk kidneys in brown trout *Salmo trutta* [10] and the proximal tubule of the nephron in the Pacific salmon [9,14] at the freshwater ontogenetic stages. To the best of our knowledge, there are no studies covering the structural changes of all the principal mesonephros functional units in salmon in both the freshwater and the seawater period of their life cycle. The Atlantic salmon *Salmo salar* Linnaeus 1758, a species with a complex intraspecific structure, is of particular interest as a research object. Populations from different rivers vary in terms of many biological characteristics, such as the duration of their freshwater period of life, their lives in seas with different salinities, and the timing of their return to their spawning grounds [1].

There are two large populations of anadromous Atlantic salmon in European Russia, and they are isolated from each other—one migrates to the Baltic Sea, the other to the Barents Sea. The allele frequencies of some important marker loci indicate a significant difference between these populations. The colonization of the Baltic region by salmon ended less than 10,000 years ago; it took place in both freshwater and saltwater periods in the history of the Baltic Sea. Currently, the foraging of the Baltic Sea’s salmon population, in contrast to the Barents Sea population, occurs in brackish waters [15]. Given these discrepancies, the aim of this article is to present the data on the mesonephros histology and ultrastructure in *Salmo salar* parr, smolts, and mature individuals from sea habitats and spawning grounds from the populations of the Baltic Sea and Barents Sea basins to contribute to the comparative evolutionary studies of kidney structure in lower vertebrates.

## 2. Materials and Methods

### 2.1. Ethics Statement

This study complied with the ARRIVE guidelines (Animal Research: Reporting of In Vivo Experiments) and was carried out in accordance with Directive 2010/63/EU on the protection of animals used for scientific purposes and the Russian Federation’s animal welfare laws.

### 2.2. Study Area

Sampling sites are marked in Figure 1. The Baltic Sea is an inland sea of the Atlantic Ocean, situated between the Scandinavian Peninsula and the mainland shores of northwestern Europe. The Baltic Sea is shallow; depths up to 200 m occupy 99.8% of its area. Luga Bay, a part of the Gulf of Finland in the Baltic Sea, cuts into the land for 18.5 km; its area is 192.9 km^2^. The average depth of the bay is 11.4 m, with a maximum of 35 m. In summer, the water warms up to 19 °C. The salinity of the water in Luga Bay varies from 0.17‰ in the south to 6.20‰ in the north. The bay receives the Luga River, which flows for 353 km and has a total drainage basin of 13,200 km^2^. River waters belong to the bicarbonate class; total dissolved-solids concentrations vary from 120–200 mg/L during high water in spring to 300–400 mg/L during baseflow. In its upper course, the Luga River flows in low, partially paludified banks. The middle and lower Luga River flow through hilly or flat terrain [16,17].

The Barents Sea, a marginal sea in the Arctic Ocean, washes the shores of Norway and Russia. The influx of warm Atlantic waters determines the temperature in the southwestern part of the sea. Here, in summer, the water on the surface warms up to 7–9 °C; in the north, the water temperature on the surface is 4–0 °C. The salinity of the surface water in the open sea ranges from 32‰ in the north to 35‰ in the southwest. In the southern part, the Barents Sea receives the Kola and Tuloma Rivers, mountainous rivers that drain into Kola Bay. The Kola River, 83 km long with a drainage basin of 3850 km^2^, has numerous rapids. The Tuloma River, 59.8 km long, with a catchment area of 21,140 km^2^, is a reservoir for almost its entire length, elongated, with a maximum width of 1.6 km and an average depth of 8 m. There are two hydroelectric power stations, one of which is equipped with a fish pass, where permanent observation of the passage of spawning and migrating Atlantic salmon has been conducted since 1958. There are many bottom and riverside springs in the Tuloma River; therefore, the river warms up poorly in summer. The waters of the Kola and Tuloma Rivers belong to the bicarbonate class; total dissolved-solids concentrations vary up to 200 mg/L [17,18].

### 2.3. Fish and Sampling

This study was performed in compliance with local laws (permission no. 78 2017 03 3049, dated 11 April 2017; no. 78 2018 03 2833, dated 21 March 2018; no. 51 2017 03 0053, dated 2 May 2018; and no. 51 2018 03 0125, dated 17 May 2018). Atlantic salmon were collected within the framework of the studies of Berg State Research Institute on Lake and River Fisheries and Nikolai M. Knipovich Polar Research Institute of Marine Fisheries and Oceanography.

Salmon from the Baltic Sea population were collected in 2017 and 2018. Parr were caught at the Luga River sampling site (altitude—0 m) in October using gill nets; smolts were sampled there in May using a floating trap net. Mature fish were collected in October using gill nets in Luga Bay in the Gulf of Finland, 14 km from the mouth, during their spawning migration (hereafter referred to as “adults”) and on spawning grounds (altitude—0 m) in the Luga River (hereafter referred to as “spawners”). Parr from the Barents Sea population were collected at the Tuloma River sampling site (altitude—0 m) by using gill nets in September 2018; smolts were sampled there in June 2018 using a floating trap net. Mature salmon were sampled using gill nets in June–September 2017 and 2018. In the Barents Sea, feeding individuals (hereafter referred to as “adults S”) were sampled, in the Tuloma River mouth, adults returning to their river of origin (hereafter referred to as “adults M”) were sampled, and on spawning grounds in the rivers Tuloma (altitude—2 m) and Kola (altitude—5 m), adults ready to spawn were sampled (hereafter referred to as “spawners”).

Each individual, one after the other, was immobilized by a stunning blow to the head, quickly measured in fork length (FL), and subjected to cervical transection by an experienced person. Subsequently, each fish was dissected, and sex was determined visually. The samples of salmon from the Baltic Sea population were as follows: parr—10 males (FL 14–18 cm) and five females (FL 17–19 cm); smolts—12 females (FL 15–19 cm); adults—six females (FL 56–60 cm); and spawners—five females (FL 58–60 cm). The samples of salmon from the Barents Sea population were as follows: parr—13 males (FL 13–16 cm) and eight females (FL 13–14 cm); smolts—five females (FL 16–20 cm); adults S—three males (FL 71–79 cm) and four females (FL 54–58 cm); adults M—five males (FL 59–69 cm); and spawners—five females (FL 58–68 cm).

Two portions of mesonephros midsection were collected from each fish. Samples were fixed in 2.5% glutaraldehyde (batch #111-30-8, Electron Microscopy Sciences, Hatfield, UK) in 0.1-M phosphate buffer, postfixed in 1% osmium tetroxide (19100, Electron Microscopy Sciences, Hatfield, UK) for 1 h at 20 ± 2 °C, dehydrated in graded acetone (013120, Himtek, Yaroslavl, Russia) and propylene oxide (13940, Electron Microscopy Sciences, Hatfield, UK), and embedded in Araldite (13940, Electron Microscopy Sciences, Hatfield, UK) [19].

### 2.4. Trunk-Kidney Histology

Semithin sections (2–3 μm) of the obtained samples were cut with a UMTP-3 microtome (Ameqs, Moscow, Russia). Ten semithin sections from each of the two portions of the kidney were prepared and stained with methylene blue. A total of 1620 sections were made. A digital image was taken for each section using a Micromed-6 light microscope (LOMO, Saint Petersburg, Russia) and processed using the Image Tool 3.0 software. These pictures were used to measure the outer diameters of the blood vessels, renal corpuscles, and tubules. The renal interstitium area was calculated as the difference between the total area of the section and the sum of the areas of blood vessels, renal corpuscles, and tubules, and reported as a percentage value [19]. From each fish specimen, 20 measurements per studied structure were taken.

### 2.5. Trunk-Kidney Ultrastructure

Ultrathin sections were prepared using five sections from each of the two portions of the kidneys using a Leica EM UC7 Ultramicrotome (Leica Microsystems, Wetzlar, Germany). A total of 810 sections were made, stained with uranyl acetate and lead citrate, and analyzed using a JEM 1011 transmission-electron microscope (JEOL, Tokyo, Japan), accelerating voltage 80 kV. For each ultrathin section, a digital image was taken, and areas of cells, organelles, and inclusions, lengths of endocytosis zone and brush border, and diameters of cilia and microvilli were measured using the Image Tool 3.0 software. Ten measurements per studied structure were obtained from each fish specimen. The numbers of mitochondria, specific granules, vesicles, and secretory granules were counted using the digital images of cells.

### 2.6. Statistical Analysis

To determine the differences between fish groups, statistical analysis was performed in two steps. First, the mean and the standard error of the mean (SEM) were calculated for the data replicates obtained for each individual. As no statistically significant differences were found between males and females, these data were combined and analyzed together in the next step. At this point, the mean ± SEM was calculated for a set of individuals of one ontogenetic stage (parr, smolts, adults, or spawners). To analyze data sets, the Kruskal–Wallis ANOVA with post hoc Wilcoxon–Mann–Whitney test (*p* < 0.05) was employed using the Statgraphics Plus software.

## 3. Results

### 3.1. Trunk-Kidney Histology

The renal interstitium is formed of hematopoietic tissue surrounding nephrons and small blood vessels; it also fills small islands located closer to the lateral side of the organ. The proportions of the renal interstitium in the total area of the kidney in the salmon from the Baltic Sea and Barents Sea populations were 48–51% (Table 1) and 50–68% (Table 2), respectively. In the Barents Sea population, the area of the renal interstitium significantly increased in adults S compared with the parr and smolts. During spawning migration, when the fish entered the mouth of the Tuloma River, the interstitium area started to decrease and, by the time of the spawning, it lowered to the values registered in the smolts (Table 2). The outer-diameter values of the blood vessels in the salmon from both populations and at all the studied ontogenetic stages showed no significant differences (Table 1 and Table 2) (Figure 2a–f).

The nephron, a functional unit of the kidney, consists of a renal corpuscle and renal tubules. In the salmon from the Baltic Sea population, the sizes of the diameters of the renal corpuscles were ranked as follows, in ascending order: smolts, adults, and parr. In the spawners, the diameters were close to those of the parr (Table 1). The Barents Sea population demonstrated a similar trend. The renal corpuscle diameter was smaller in the adults S, smolts, and adults M in comparison with the parr and spawners (Table 2).

The renal corpuscle is followed by a tubule, which is subdivided into the proximal and distal sections. In the Baltic Sea population, the outer diameters of the proximal tubules was very similar in the parr, adults, and spawners, and they were significantly smaller in the smolts (Table 1). In the Barents Sea population, the outer diameters of the proximal tubules decreased from the parr to the smolts to the adults S, followed by an increase from the adults M to the spawners (Table 2). The outer-diameter values of the distal tubules in the parr and spawners from the Baltic Sea population were larger than those in the smolts and adults (Table 1). The same trend was observed in the Barents Sea population; the outer diameter of the distal tubules in the parr was also the largest; it subsequently decreased from the smolts to the adults S and started to increase again from the adults M to the spawners (Table 2) (Figure 2a–f).

### 3.2. Trunk-Kidney Ultrastructure

The renal corpuscle, a structural unit of the nephron, consists of a tuft of capillaries (glomerular capillaries) and Bowman’s capsule (Figure 3c). The renal corpuscles were structured similarly in the salmon from both populations at each studied ontogenetic stage. The Bowman’s capsule has two layers, parietal and visceral, forming the urinary space (Bowman’s space), which, in the Baltic Sea population, increased from the adults and smolts to the spawners to the parr (Table 3). In the Barents Sea population, a similar pattern was observed, with the width of the urinary space in the adults M close to that in the spawners (Table 4).

The parietal layer of Bowman’s capsule is formed by a basement membrane lined with squamous epithelial cells with centrally located nuclei. The visceral layer consists of podocytes lying on the basement membrane (Table 3 and Table 4). In the podocyte cytoplasm in the salmon from the Barents Sea population, a well-developed rough endoplasmic reticulum and numerous ribosomes and mitochondria occupying almost the entire cell volume were found (Figure 3a); a quite different picture was observed in the Baltic Sea population—a less developed endoplasmic reticulum, and solitary ribosomes and mitochondria (Figure 3b).

The basement membrane forms the supporting structure for podocytes and capillaries (Figure 3a,b). In the Baltic Sea population, the width of the basement membrane in the parr and spawners was significantly lower than that in the adults and smolts (Table 3). In the Barents Sea population, the thickening of the basement membrane occurred, in ascending order, in the parr, followed by the spawners and smolts, followed by the adults M and the adults S (Table 4). Beneath the basement membrane lie the glomerular capillaries. In both populations, the smolts had glomerular capillaries that were significantly smaller in diameter; in the adults S from the Barents Sea population, the capillary diameters were closer to those of the smolts than to those of the parr and the spawners, unlike in the adults from the Baltic Sea (Table 3 and Table 4).

In the studied salmon, the proximal tubules had epithelial cells that were typical of this unit of the nephron. The proximal tubules are distinguished from the distal tubule cells by the presence of a brush border. The ultrastructure analysis of the proximal tubules showed two types of epithelial cell. The type I epithelial cells were located at the beginning of the proximal tubule. These were the largest cells, elongated and pyramidal in shape, with tightly stacked smooth endoplasmic reticula. In the Baltic Sea population, the areas and heights of the type I epithelial cells were significantly greater in the parr and spawners than in the smolts and spawners (Table 5). The same pattern was observed in the salmon from the Barents Sea population, except that the cell areas in the adults M were very close to those in the spawners (Table 6).

These cells had spherical nuclei located in the basal parts, where numerous curls of the smooth endoplasmic reticulum were observed (Figure 3d,f). The least developed smooth endoplasmic reticula were found in the parr and spawners from both of the studied populations (Figure 3d). Numerous strands of smooth endoplasmic-reticulum tubules were observed in the cells of the smolts, adults, adults S, and adults M; in those of the adults S, many loops were formed (Figure 3f). The widths of the cisternae of the smooth endoplasmic reticulua were quite similar between the studied populations, with the only difference shown in those of the adults from the seawater habitats, as the Barents Sea salmon had significantly larger cisternae when living at sea (Table 5 and Table 6). The cytoplasm of the type 1 epithelial cells was packed with mitochondria. In both the studied populations, the area and number of mitochondria were significantly smaller in the parr and spawners than in the smolts and adults (Table 5 and Table 6).

The type 1 epithelial cells contained lysosomes and large electron-dense secretory granules (Figure 3g). The number of secretory granules changed dramatically across the studied ontogenetic stages. In the Baltic Sea population, the number increased in the following order: from parr to smolts to adults. An insignificant decrease was observed in the spawners, whose granules were three times smaller (Table 5). In the Barents Sea population, the number of secretory granules changed in a similar pattern, except that the decrease from the adults to the spawners was more pronounced; the same three-times reduction in granule area was registered in the adults M and spawners (Table 6). The apical parts of the cells, adjacent to the brush border, were observed to have well-developed zones of endocytosis. These zones had tubulovesicular networks. The lengths of the endocytosis zones had no significant differences across the ontogenetic stages of the salmon from both populations (Table 5 and Table 6). The brush borders were formed by numerous microvilli facing the tubule lumen (Figure 3e,g). No pronounced changes were registered in the brush-border length in the salmon from the Baltic Sea population (Table 5), while in the Barents Sea population, the brusher borders were significantly shorter in the adults S (Table 6). The microvillus diameter did not change in this population (Table 6); in the Baltic Sea population, the parr had narrower microvilli (Table 5). Among the type I epithelial cells in the nephrons of the salmon at each ontogenetic stage, single cells with a brush border formed by cilia were found; the cilia diameters were constant at every ontogenetic stage in both populations (Table 5 and Table 6).

The type II epithelial cells, located at the ends of the proximal tubules, were similar to the type I epithelial cells in general (Figure 4a–f). In the Baltic Sea population, the cell area and height increased in the following order: from the smolts to the adults to the parr and spawners (Table 7). In the Barents Sea population, these parameters were smallest in the adults S and greatest in the parr and spawners (Table 8). The type II epithelial cells had spherical nuclei located in their basal parts; there were no statistically significant differences in nucleus area between the studied salmon groups (Table 7 and Table 8). Mitochondria were found in the compartments of the basal labyrinth (Figure 4a,c,e). The area of mitochondria increased in the following order in both populations: from the parr to the spawners to the smolts to the adults (Table 7 and Table 8). The number of mitochondria rose gradually from the spawners and parr to the adults to the smolts in the Baltic Sea population (Table 7). In the Barents Sea population, significant differences in the number of mitochondria were observed between the parr, adults M, and spawners on the one hand and the smolts and adults S on the other (Table 8).

In the Baltic Sea population, the smooth endoplasmic reticulum in the cytoplasm of type II epithelial cells was least developed in the parr and spawners; separate strands of tubules stretched along the cells enveloping the mitochondria (Figure 4a). Numerous strands of tubules of the smooth endoplasmic reticulum were found in the cells of the smolts and adults (Figure 4c). Similar ultrastructural features were observed in the Barents Sea population; solitary strands of tubules of the smooth endoplasmic reticulum were found in the parr and spawners, while in the cells of the smolts and adults M, the strands of the tubules became numerous. In the adults S, folds of the smooth endoplasmic reticulum stretched in the cytoplasm and formed complex interlacements (Figure 4e). A characteristic feature of type II epithelial cells is the absence of secretory granules in the cytoplasm. The lengths of the endocytosis zone had no pronounced differences between the studied salmon, except that, in the Barents Sea population, they were greater in the adults S compared with the salmon at the other ontogenetic stages (Table 8). The brush-border lengths and the diameters of the microvilli and cilia did not differ significantly between the salmon at the studied ontogenetic stages from both populations (Table 7 and Table 8).

The distal tubules of the nephron were lined by the tallest and widest cells among the tubule epithelial cells. The spherical nuclei were shifted to the basal parts of the cells. In the Baltic Sea population, the smolts had the smallest cells (Table 9). In the Barents Sea population, the smallest cell areas and shortest heights were registered in the smolts and adults S (Table 10). The nucleus areas were significantly greater in the parr from both populations (Table 9 and Table 10). The folds of the smooth endoplasmic reticulum, similar to those in the epithelial cells of the proximal tubule, were better developed in the smolts than in the parr and spawners (Figure 5a–e). In the adults S, closer to the apical parts of the cells, the folds of the smooth endoplasmic reticulum formed interlacements (Figure 5f). Large elongated electron-dense mitochondria stretching along the longitudinal axis of the cells were observed in all the studied salmon. In the Baltic Sea population, the sizes of the mitochondrial area were as follows, in ascending order: from the spawners to the parr to the adults to the smolts (Table 9). In the Barents Sea population, the areas of the mitochondria were as follows, in ascending order: from the spawners to the adults M and parr to the smolts to the adults S (Table 10). The number of mitochondria in the salmon from the Baltic Sea population was the greatest in the smolts (Table 9). In the Barents Sea population, the numbers of mitochondria in the smolts and adults S exceeded those in the parr, adults M, and spawners (Table 10). There were no zones of endocytosis in the epithelial cells of distal tubules. In the apical parts of the cells, lobed cytoplasmic processes facing the lumen of the tubule were observed (Figure 5b,d,f). In the smolts and adults from the seawater habitats, the cytoplasm of the distal-tubule epithelial cells contained solitary mitochondria, vesicles, and separate reticulum tubules (Figure 5d,f).

At the ultrastructural level, in the renal interstitium, leukocytes at various maturity stages were found in the salmon at all the studied ontogenetic stages from both populations. In addition, chloride cells were registered in the renal interstitia of the smolts, adults, and spawners, and cells with radially arranged vesicles were found in the smolts and spawners from both populations and in the parr from the Barents Sea population.

Lymphocytes are the smallest spherical cells among leukocytes; in the salmon, the cell areas varied in a range of 17.9–32.5 μm^2^ (Table 11 and Table 12). In the Baltic Sea population, the smallest lymphocytes were found in the interstitia of the spawners and parr and the largest were found in the interstitia of the adults (Table 11). A similar trend was observed in the salmon from the Barents Sea population (Table 12). The characteristic feature of lymphocytes is a large nucleus occupying almost the whole cell. Regardless of the ontogenetic stage, in the salmon from both populations, the nuclei of the lymphocytes were spherical. No statistically significant differences were found in the nucleus areas between the studied ontogenetic stages. The heterochromatin was less concentrated in the parr and smolts than in the adults and spawners. Numerous loose ribosomes and mitochondria filled the fine rim of the lymphocyte cytoplasm (Figure 6a–f). The cytoplasm of some cells in the smolts contained centrioles (Figure 6b). In the Baltic Sea population, in the smolts and adults, the number of mitochondria significantly exceeded that of the parr, and the area of the mitochondria was largest in the adults (Table 11). In the mitochondria of the adults, cristae were the most developed (Figure 6c). In the Barents Sea population, the number of mitochondria in the cytoplasm of the smolts and adults S was twice that in the cytoplasm of the parr; the area of the mitochondria was largest in the adults S and adults M (Table 12). The mitochondria with the most well-developed cristae were found in the lymphocytes of the adults S and adults M (Figure 6d,e). The lymphocyte cytoplasm in the spawners from the Baltic Sea population and in the adults M and the spawners from the Barents Sea population contained many separate rough and smooth endoplasmic reticulum cisternae (Figure 6e,f).

The vast majority of the plasma cells in the studied salmon were oval. In the Baltic Sea population, the largest area of plasma cells was registered in the adults; it significantly exceeded those in the spawners and parr (Table 11). No significant differences in cell area were found between the studied ontogenetic stages of salmon from the Barents Sea population, although the values tended to show a similar pattern (Table 12). The plasma cells had eccentrically shifted spherical nuclei. Most of the heterochromatin adjoined the inner membrane of the nuclear envelope. Furthermore, separate clods were located evenly across the sections of the nuclei. The plasma-cell cytoplasm contained stacks of well-developed rough endoplasmic reticulum, loose ribosomes, electron-dense lysosomes, and mitochondria. According to the results of the electron-microscope analysis, the rough endoplasmic reticulum development in the plasma cells depended on the ontogenetic stage of the salmon (Figure 7a–d). In the plasma cells of the salmon from both the studied populations, the widths of the rough endoplasmic reticulum cisternae increased from 0.13 ± 0.01 in the parr to 0.16 ± 0.00 μm in the smolts (Figure 7a). In the adults from the Baltic Sea and the adults S from the Barents Sea population, cisternae as wide as 0.28 ± 0.02 μm were found in most of the plasma cells (Figure 7b). However, in the cells of the adults M and spawners from the Barents Sea population and the spawners from the Baltic Sea population, cisternae of 0.38 ± 0.02 μm occupied the entire area of the cells (Figure 7c,d). The largest numbers of mitochondria in the plasma cells were registered in the smolts and adults from both the studied populations; in the parr and spawners, the numbers were significantly lower (Table 11 and Table 12). There were no pronounced differences mitochondrial area between the parr, smolts, and spawners from both populations; the mitochondrial areas were significantly greater in the adults from the sea habitats (Table 11 and Table 12).

The macrophages are the largest agranulocytes (Table 11 and Table 12), spherical, with an eccentrically located nucleus, regardless of the ontogenetic stage of the salmon (Figure 7e,f). In both populations, the area of macrophages was smallest in the spawners and largest in the adults from the sea habitats (Table 11 and Table 12). Most of the heterochromatin was located on the periphery of cells, adjoining the inner membrane of the nuclear envelope. In the central part of the nuclei, the heterochromatin was in the form of clods or threads. The cytoplasm was packed with phagosomes that occupied almost the entire cell volume. The cytoplasm of most of the macrophages in the mesonephros of the spawners from both populations contained dead cells, presumably lymphocytes (Figure 7f). The cytoplasm of the macrophages also contained mitochondria, short sheets of rough endoplasmic reticulum, loose ribosomes, lysosomes, and small light vesicles (Figure 7e,f). In both populations, the mitochondria in the smolts and adults were larger and more abundant than those in the parr and spawners (Table 11 and Table 12). In the parr from both populations, the cytoplasm of the macrophages contained electron-dense pigment granules (Figure 7e).

Neutrophils are spherical cells with eccentrically shifted nuclei. In the Baltic Sea population, the sizes of the cell areas of the neutrophils in the renal interstitia increased from the parr to the spawners (Table 13), while in the Barents Sea population, the cell area increased from the parr to the smolts to the adults S and then decreased in the adults M and the spawners (Table 14). In both populations, the majority of the neutrophils observed in the parr were metamyelocytes; some of the cells were banded neutrophils. Most of the cells in the smolts, adults, adults S, and adults M were banded neutrophils. In the spawners from both populations, banded and segmented neutrophils were present in equal volumes. Almost every segmented neutrophil had a nucleus with two sections. The heterochromatin was cloddy, located mainly on the cell periphery, adjoining the inner membrane of the nuclear envelope; separate clods were found in the central parts of the nucleus (Figure 8a–f). In the Baltic Sea population, a dramatic increase in the average area of the mitochondria in the neutrophils was registered from the parr to the spawners (Table 13). The number of mitochondria in the parr was smaller than those in the salmon at the other ontogenetic stages (Table 13). In the Barents Sea population, the area of mitochondria increased sharply from the parr to the smolts to the adults S, followed by a symmetrical drop to the adults M and then to the spawners (Table 14). The numbers of mitochondria in the neutrophils of the parr and spawners were similar and significantly smaller than those of the smolts, adults S, and adults M (Table 14). The salmon at all the studied ontogenetic stages from both populations had neutrophils containing specific granules of two types: (1) electron-dense with fibrils located along the granules; and (2) granules that were lighter in electrons in the central part, with dark fibrillar edges (Figure 8a–f). In the salmon from both the studied populations, the numbers of specific granules in the neutrophils increased from the parr to the spawners (Table 13 and Table 14).

Eosinophils, a type of granulocyte containing spherical electron-dense granules that are homogenous in structure, were observed in the salmon at all the studied ontogenetic stages from both populations (Figure 8g–i). In the Baltic Sea population, the number of granules in the eosinophils in the parr was significantly smaller than those in the salmon at all the other ontogenetic stages (Table 13); the same pattern was registered in the Barents Sea population (Table 14). Furthermore, the differences in the average area of the granules among the ontogenetic stages, although statistically significant, were not highly pronounced in the salmon from the Baltic Sea population (Table 13). Additionally, in the Barents Sea population, the average area of the eosinophil granules was more than three times greater in the spawners than in the salmon at the other studied ontogenetic stages (Table 14). Eosinophils, which are spherical cells, had eccentrically located nuclei in all the studied groups of salmon. Most of the eosinophils in the parr were immature cells, with spherical nuclei; only 10% were banded cells. An equal amount of banded and segmented eosinophils was registered in the smolts and spawners. All the segmented cells had bi-lobed nuclei. Cloddy heterochromatin was found mainly on the cell periphery, adjoining the inner membrane of the nuclear envelope; separate clods were found in the central parts of the nuclei (Figure 8g–i). In the eosinophils of the salmon from the Baltic Sea population, the area and number of mitochondria were smallest in the parr (Table 13). The average area of the mitochondria in the salmon from the Barents Sea population increased from the parr to the smolts to the adults S and then decreased abruptly in the adults M and spawners; the change in the number of mitochondria demonstrated a similar this pattern (Table 14). In the Barents Sea population, the eosinophil cytoplasm of the adults M and spawners contained large phagosomes (Figure 7i).

Cells with radially arranged vesicles are spherical or trihedral in shape. Almost all the cells with radially arranged vesicles in the renal interstitia of the studied salmon were spherical, with eccentrically located spherical nuclei (Figure 9c). In all the salmon, the heterochromatin of the cells was very condensed (Figure 9c–e). The cytoplasm was packed with short stacks of smooth endoplasmic reticulum and mitochondria. There were vesicles and mitochondria in the apical parts of the cells (Figure 9a–e). Cells with radially arranged vesicles were found in the smolts and spawners from the Baltic Sea population; statistically significant differences were registered in the areas and numbers of mitochondria, which were greater in the smolts (Table 15). These cells were also observed in the parr from the Barents Sea population; differences were found between the areas and the numbers of mitochondria and vesicles. The area and number of mitochondria and vesicles were greater in the smolts than in the parr (Table 16). Vesicles with fibrillar structures were found in the cytoplasm of these cells in the salmon from both populations at the aforementioned ontogenetic stages (Figure 9a,b,e). In the smolts and spawners, cells with electron-transparent vesicles were also observed (Figure 9c,d).

Chloride cells were found in the renal interstitia of the salmon at all the studied ontogenetic stages, except for the parr. Long and narrow (Figure 9f), these cells were diffusely scattered in the kidney parenchyma and also surrounded the blood vessels and nephron tubules. The spherical nuclei were located closer to the peripheries of the cells; the heterochromatin was slightly condensed. The cytoplasm was electron-light with numerous mitochondria. In the salmon from the Baltic Sea population, the largest areas of cells, nuclei, and mitochondria were registered in the adults (Table 15). The numbers of mitochondria were as follows, in ascending order: from the spawners to the smolts to the adults (Table 15). In the Barents Sea population, the areas of the cells and nuclei and the areas and numbers of mitochondria were significantly greater in the adults S compared with those in the salmon at the other ontogenetic stages (Table 16). The chloride cells contained electron-dense granules and tubular endoplasmic reticula; single tubules were distributed all over the cells (Figure 9f,g).

## 4. Discussion

In the course of ontogenesis, immature Atlantic salmon preparing to live in sea water and sexually mature fish performing their spawning migration undergo physiological adaptations, which, among other processes, are represented by changes in osmoregulation. Several organs are involved in osmoregulation: the kidneys, gills, digestive tract, bladder, and endocrine glands [4]. Previous studies on *Pseudetroplus maculatus* (Bloch 1795), *Alburnus escherichii* Steindachner 1897, and *Salmo trutta* showed that adaptation to a hyperosmotic environment leads to a decrease in the diameter of Bowman’s capsules, accompanied by a decrease in glomerular filtration rate, which indirectly indicates a reduction in the formation of primary urine [10,20,21]. Moreover, smolts can have epithelial cells that are smaller in area and height in their proximal tubules, which indicates a low glomerular filtration rate, a distinctive characteristic of marine fishes [8,9,22,23]. There is evidence that in coho salmon *Oncorhynchus kisutch* (Walbaum, 1792), the urine flow is eleven times weaker and the glomerular filtration six times weaker in those living in the sea than in those living in rivers by the time of spawning [24]. The diuresis of euryhaline teleosts is regulated by two main renal mechanisms: glomerular filtration and tubular reabsorption. In freshwater environments, glomerular filtration is high, and tubular reabsorption is low, while in sea water, the filtration rate is greatly reduced, and reabsorption is accelerated. One of the few relevant experimental studies showed a reduction of almost 50% in the urine flow and about the same decrease in the glomerular filtration rate in *Oncorhynchus mykiss* (Walbaum 1792) smolts compared with pre-smolts [25].

Our study shows that ultrastructural changes in the nephron, which are known to affect the glomerular filtration rate and the amount of urine produced in salmon kidneys, occurred as early as the smolting stage. Such changes in the smolt mesonephros, including reductions in the diameters of the renal corpuscle, glomerular capillaries, and renal tubule, in the urinary space width, and in the size of the proximal-tubule epithelial cells, as well as the thickening of the basement membrane, reflect fundamental alterations that may be considered pre-adaptations to a habitat shift or, in other words, as preparation for hyperosmotic conditions.

The data on the nephron histology and ultrastructure also showed that in the Barents Sea population, the adult salmon caught in the saline environments had the renal corpuscles and proximal and distal tubules with the smallest diameters, the narrowest urinary spaces, and the thickest basement membranes. These parameters in the parr and adults from the river mouths reflected a transition in the osmoregulation process, supporting their seaward or spawning migration. The less pronounced differences in these morphometric features between the salmon from the Baltic Sea population might be explained by the low salinity values in Luga Bay.

The comparison between the adults returning to their river of origin (sampled in the mouth of the Tuloma River) with the salmon at the other ontogenetic stages is of particular interest. Based on the presence of marine ectoparasites *Lepeophtheirus salmonis*, which is known to fall off the fish body during the first day on which salmon are in fresh water, these adult salmon had spent less than 24 h in the fresh water of the Tuloma River. During that first day that the fish spent in fresh water, the structural rearrangement of the nephron observed in the salmon at the “freshwater” ontogenetic stages did not yet occur completely. These results are consistent with the findings of Talbot et al. (1992) [26] on urine regulation in Atlantic salmon *Salmo salar* returning to the sea after spawning. These fish, known as kelts, have limited hypoosmoregulation ability. Talbot et al.’s study showed that freshwater-adapted kelts demonstrated a change in their osmoregulation, involving the development of the characteristics of seawater-adapted salmon, in 48 h. The urine-flow rate showed a six-time decrease in 24 h; over the same period, the urine osmolality became isosmotic with the plasma, and the secretion of Mg^2+^ by the kidney tubules increased the urine concentration by a factor of 200 [26].

The ultrastructural changes in the epithelial cells of the proximal and distal tubules in the smolts and adults caught in the Barents Sea, the mouth of the Tuloma River, and the Gulf of Finland compared with those in the parr and spawners, such as the increase in the size and number of mitochondria, the development of a smooth endoplasmic reticulum, and the extension of its cisterns, may have resulted from the increased secretion of divalent ions (Mg^2+^, Ca^2+^, SO4^2−^) and phosphates, and the reabsorption of water and sodium. Ions from the interstitial fluid enter the cell through the basal membrane, and then move towards and through the apical membrane and enter the lumen of the tubule. The smooth endoplasmic reticulum is directly related to this mechanism, since this cell structure provides the molecular basis for the operation of ion pumps [22]. In the salmon entering the mouth of the Tuloma River (and spending less than 24 h in fresh water), the structural rearrangement supporting the molecular pumps occurred only in the distal tubules. Our data are in agreement with those in the study by Beyenbach (1995), showing that changes in the environment accompanied by a pronounced decrease in the ion content, as in shifts from seawater to fresh water, allow the renal cells to function at a lower level of metabolism [27].

Katoh et al. (2008) reported that the distal tubules of *Oncorhynchus mykiss*, a euryhaline salmonid species, contain apically located Na^+^/K^+^/2Cl^−^ cotransporters, which probably play a role in Na^+^, K^+^, and 2Cl^−^ absorption, allowing the ions to be transported back into the body through basolateral-specific ion channels and/or ion-transport proteins, such as the Na^+^/K^+^-ATPase. Thus, the distal tubules can probably switch between secretion and absorption, depending on the osmoregulatory needs of the fish [28]. This proposed mechanism might be responsible for the great number of vesicles scattered in the lobed cytoplasmic processes, as well as the large mitochondria and separate endoplasmic reticulum tubules in the distal-tubule epithelial cells in the smolts and adult salmon from the sea habitats. Furthermore, the better development of the smooth endoplasmic reticulum and the increase in the number of mitochondria in the tubule cells in the adults from the Barents Sea compared with those in the adults caught in the Gulf of Finland might be explained by the fact that the salinity of water encourages the development of the cellular structures responsible for the implementation of ion transport through the membranes of tubule cells.

In both the studied populations, the adult salmon at the seawater stage of the life cycle had the largest chloride cells (ionocytes) in the mesonephros interstitium, as well as the highest number of mitochondria in these cells, which supports previous evidence of their involvement in ion transport [4,29]. Since chloride cells were not found in the parr from either population, these cells are apparently formed later in the ontogenetic process, at the pre-smolt or smolt stage.

According to the results of the leukocyte-ultrastructure analysis, the cell areas of almost all the leukocytes in the mesonephros of the adults from the seawater habitats exceeded those in the parr and adults heading to or having reached the spawning grounds. In all the leukocytes, the number of mitochondria started to increase during smolting. In both studied populations, the sizes of the lymphocytes and plasma cells and the sizes and numbers of their mitochondria seemed to be closely associated with the preparation for the salinity shift, as well as the shift itself.

Mitochondria are key organelles in intermediate cellular metabolism and energy conversion. The latter occurs at the inner mitochondrial membrane, which can be subdivided into the peripheral or “boundary membrane” and the cristae, which are invaginations into the matrix separated from the peripheral region by narrow tubular junctions [30]. The mitochondrial inner membrane unfolds to accommodate a larger matrix volume or increasing folds in response to matrix shrinkage; membrane folding and unfolding are linked to the energetic state through changes in the architecture of the cristae, which contain respiratory-chain supercomplexes, thus determining mitochondrial respiratory efficiency [30,31]. Curiously, in our study, the lymphocytes of the salmon living in salt water had the most developed cristae and, in addition, the largest mitochondria and cell areas. Apart from the interrelations between the sizes, this observation is interesting in terms of mitochondrial functions. It is a common notion that cristae provide surface areas on which chemical reactions to occur; therefore, the development of these structures in interstitial cells is likely to be a reflection of the metabolic rate in fish.

Changes in mitochondrial matrix volume are associated with a range of functional responses. Mitochondrial volume is coordinated by transmembrane ion and substrate transporters in response to cellular energy demand and proliferation, division, or differentiation [31]. There are findings supporting the hypothesis that an increase in matrix volume can activate the respiratory chain, thereby increasing ATP production [32]. This explanation for of the larger mitochondria in the interstitial cells of the adult salmon seems adequate, as high ATP demands would be necessary for efficient protein synthesis in all the systems of the fish body to support somatic growth and gonadal development. However, this might not be the main reason, as mitochondria are known to be directly involved in adaptive mechanisms to high salinity [33].

Taken together, the structural changes in the interstitial cells, including the development of the rough endoplasmic reticulum in the lymphocytes and plasma cells, the increase in specific granules in the neutrophils and eosinophils, and the increasing abundance of mature forms of granulocytes in the kidney interstitia, indicate that the increased activation of cellular immunity, which is known to occur under hyperosmotic conditions, started during the parr–smolt transformation. The occurrence of dead lymphocytes in the macrophage phagosomes, large numbers of specific granules in neutrophils and eosinophils, and a well-developed rough endoplasmic reticulum in the agranulocytes are known as indications of the increased activation of cellular immunity in salmon heading to their spawning grounds [34]. Similar characteristic changes were registered in the spawners of brown trout *Salmo trutta* [10]. Another sign of cellular-immunity activation was the large number of cells with radially arranged vesicles in the kidney interstitia of the smolts and spawners. As a point of interest, the adults from the sea and river-mouth habitats did not possess these kinds of cell, nor did the parr, except for three solitary cells found in two parr individuals from the Barents Sea population. The ultrastructures of the cells with radially arranged vesicles strongly resembled the nonspecific or natural cytotoxic cells described in teleosts [35]. Nonspecific cytotoxic cells, which are a type of natural-killer-cell homolog, are responsible for nonspecific cell-mediated cytotoxicity mechanisms in fish; they are known to act on various target cells, including virus-infected cells, tumor cells, and protozoan parasites, and they may also be involved in antibacterial immunity by triggering the production and secretion of cytokines [36].

These differences between the studied ontogenetic stages indicate the extremely low rate of proliferation of these cells in the parr and adult salmon living in saline environments. Although these cells were not observed in the adults, based on the changes observed in the leukocytes, apparently, the activation of cellular immunity had already occurred when the salmon entered the rivers on their way to their spawning grounds.

## 5. Conclusions

In salmon at different ontogenetic stages, pronounced structural rearrangements were registered both in the nephron and in the renal interstitium. Regardless of the cell type, there were changes in the areas of the cells and the number and structure of organelles responsible for transport, synthesis, energy production, etc. Most of the changes occurred at the smolting stage; therefore, they can be considered cytological markers of the preparation for habitat shift during seaward migration. In the Baltic Sea population, the less pronounced differences in the morphometric characteristics of the nephron between the studied ontogenetic stages of salmon may be explained by the low salinity in Luga Bay.

## Figures and Tables

**Figure 1 biology-12-00750-f001:**
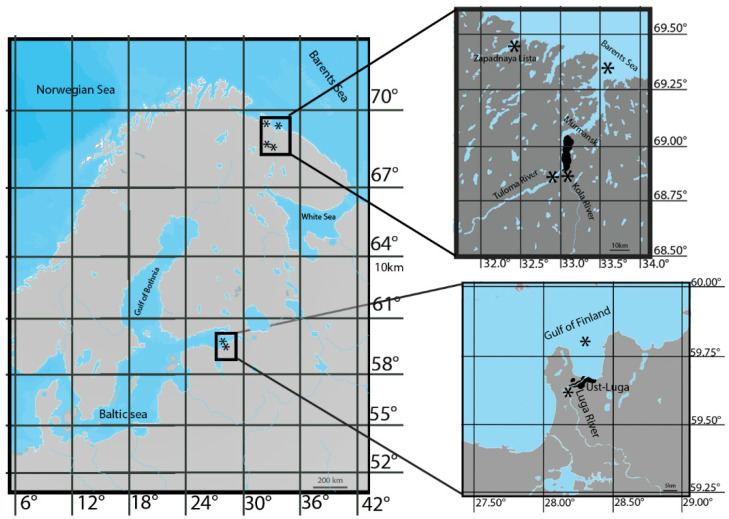
Location of the study areas. Asterisks mark the sampling sites; black-colored areas mark urban territories of Murmansk city and Ust-Luga city.

**Figure 2 biology-12-00750-f002:**
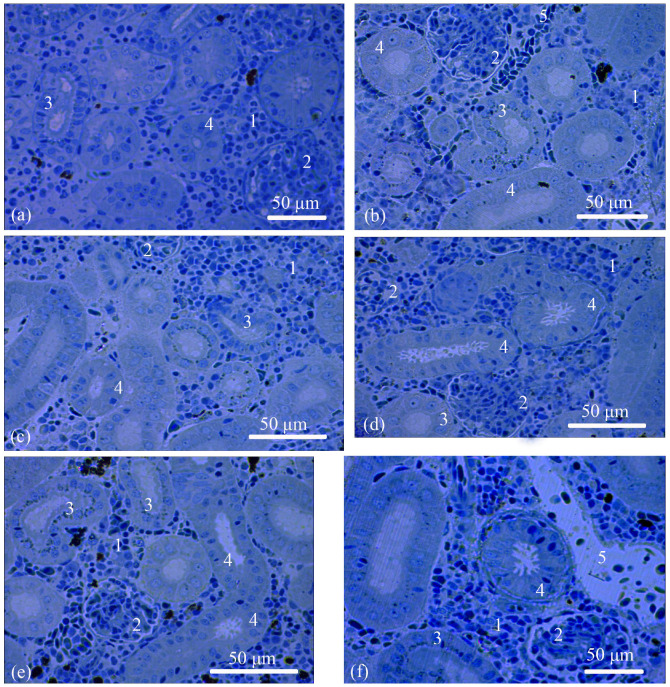
Histologies of the trunk kidneys: parr from the Baltic Sea population (**a**); smolt from the Baltic Sea population (**b**); smolt from the Barents Sea population (**c**); adult from the Baltic Sea population (**d**); adult S from the Barents Sea population (**e**); and spawner from the Barents Sea population (**f**). Renal interstitium (1), renal corpuscle (2), proximal tubule (3), distal tubule (4), and blood vessel (5).

**Figure 3 biology-12-00750-f003:**
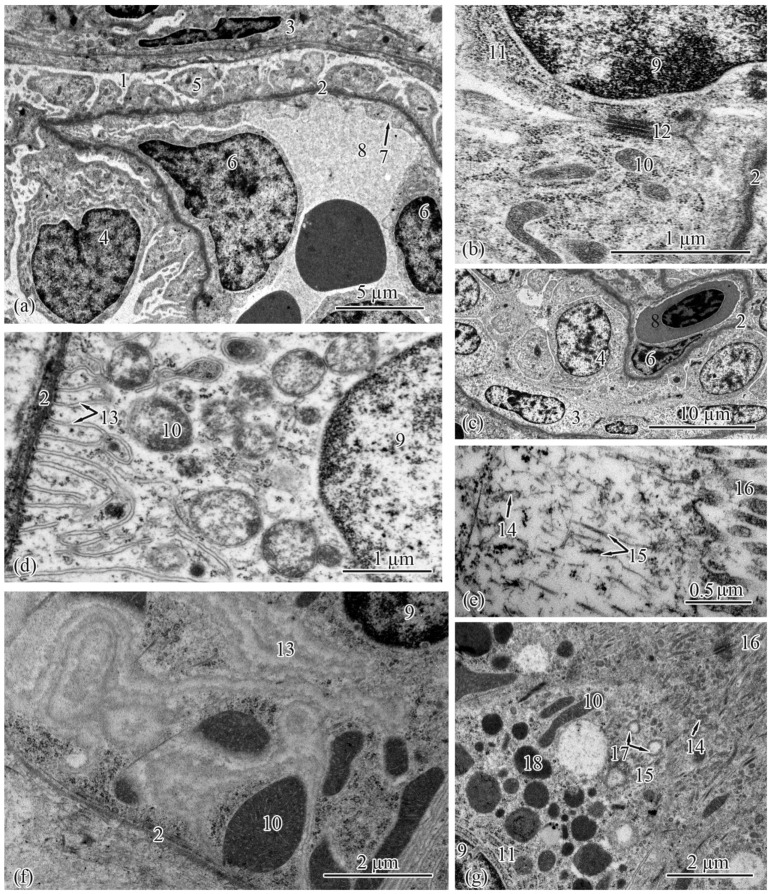
Ultrastructure of the renal corpuscle and type I epithelial cells of the proximal tubule in the trunk kidneys of the Atlantic salmon. Part of a renal corpuscle in a parr from the Barents Sea population (**a**); podocyte in an adult from the Baltic Sea population (**b**); part of a renal corpuscle in an adult S from the Barents Sea population (**c**); basal part of type I epithelial cell of the proximal tubule in a parr from the Barents Sea population (**d**); endocytosis zone and a portion of the brush border in a parr from the Baltic Sea population (**e**); basal part of type I epithelial cell of the proximal tubule in an adult S from the Barents Sea population (**f**); endocytosis zone and a portion of the brush border in an adult S from the Barents Sea population (**g**). Cell units: urinary space (1), basement membrane (2), squamous epithelial cell (3), podocytes (4), cytopodia (5), endothelial cell (6), fenestra endothelial cells (7), capillary (8), nucleus (9), mitochondria (10), rough endoplasmic reticulum (11), gap junction (12), smooth endoplasmic reticulum (13), tubular endoplasmic reticulum (14), microfibrils (15), brush border microvilli (16), vesicle (17), secretory granules (18).

**Figure 4 biology-12-00750-f004:**
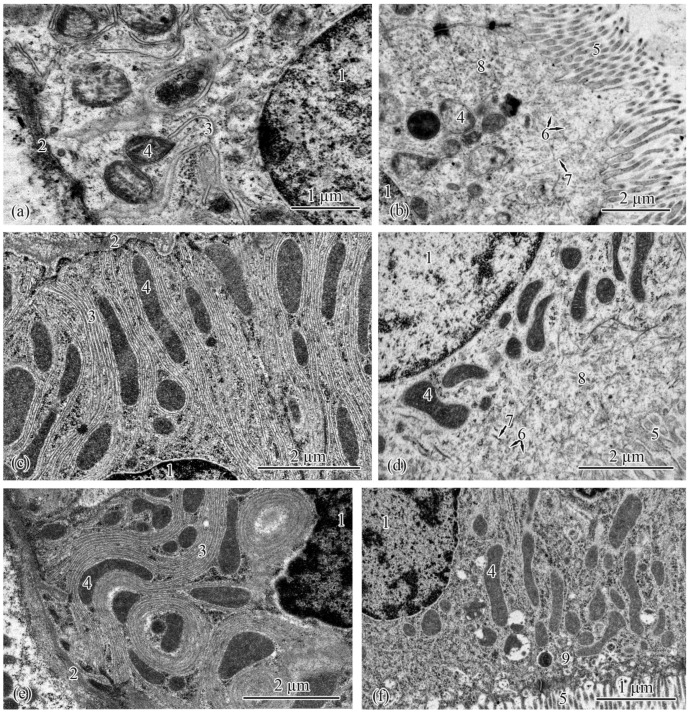
Ultrastructure of type II epithelial cells of the proximal tubule in the trunk kidneys of Atlantic salmon. Basal part of type II epithelial cell of the proximal tubule in a parr from the Baltic Sea population (**a**); endocytosis zone and a portion of the brush border in a parr from the Baltic Sea population (**b**); basal part of type II epithelial cell of the proximal tubule in an adult from the Baltic Sea population (**c**); apical part of type II epithelial cell of the proximal tubule in an adult from the Baltic Sea population (**d**); basal part of type II epithelial cell of the proximal tubule in an adult S from the Barents Sea population (**e**); apical part of type II epithelial cell of the proximal tubule in an adult S from the Barents Sea population (**f**). Cell units: nucleus (1), basement membrane (2), smooth endoplasmic reticulum (3), mitochondria (4), brush border microvilli (5), vesicle (6), microfibrils (7), tubular endoplasmic reticulum (8), endocytosis zone (9).

**Figure 5 biology-12-00750-f005:**
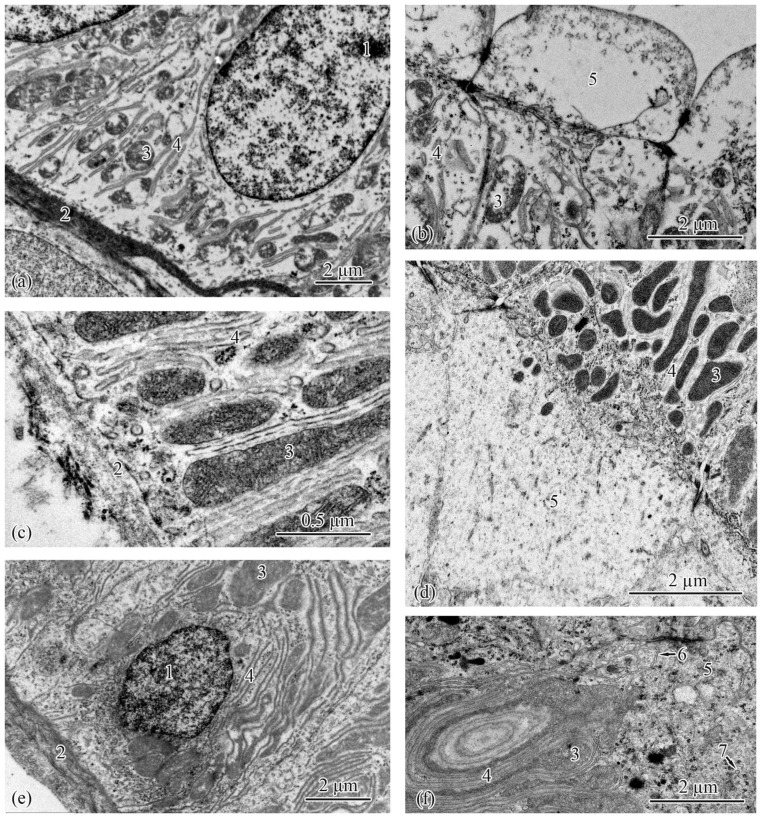
Ultrastructure of the distal tubule in the trunk kidneys of Atlantic salmon. Basal part of the epithelial cell of the distal tubule in a parr from the Baltic Sea population (**a**); apical part of the epithelial cell of the distal tubule in a parr from the Baltic Sea population (**b**); basal part of the epithelial cell of the distal tubule in an adult from the Baltic Sea population (**c**); apical part of the epithelial cell of the distal tubule in an adult from the Baltic Sea population (**d**); basal part of the epithelial cell of the distal tubule in an adult S from the Barents Sea population (**e**); apical part of the epithelial cell of the distal tubule in an adult S from the Barents Sea population (**f**). Cell units: nucleus (1), basement membrane (2), mitochondria (3), smooth endoplasmic reticulum (4), lobed cytoplasmic process (5), tubular endoplasmic reticulum (6), vesicle (7).

**Figure 6 biology-12-00750-f006:**
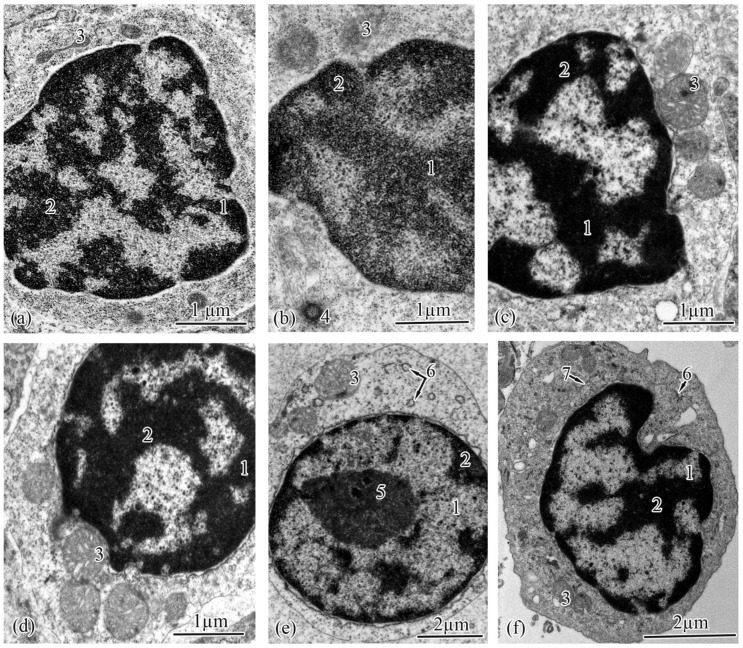
Ultrastructure of lymphocytes in the renal interstitia of Atlantic salmon. Lymphocyte in a parr from the Baltic Sea population (**a**); lymphocyte in a smolt from the Baltic Sea population (**b**), lymphocyte in a spawner from the Baltic Sea population (**c**); lymphocyte in an adult S from the Barents Sea population (**d**); lymphocyte in an adult M from the Barents Sea population (**e**); lymphocyte in a spawner from the Barents Sea population (**f**). Cell units: nucleus (1), heterochromatin (2), mitochondria (3), centrioles (4), nucleolus (5), cisternae of rough endoplasmic reticulum (6), cisternae of smooth endoplasmic reticulum (7).

**Figure 7 biology-12-00750-f007:**
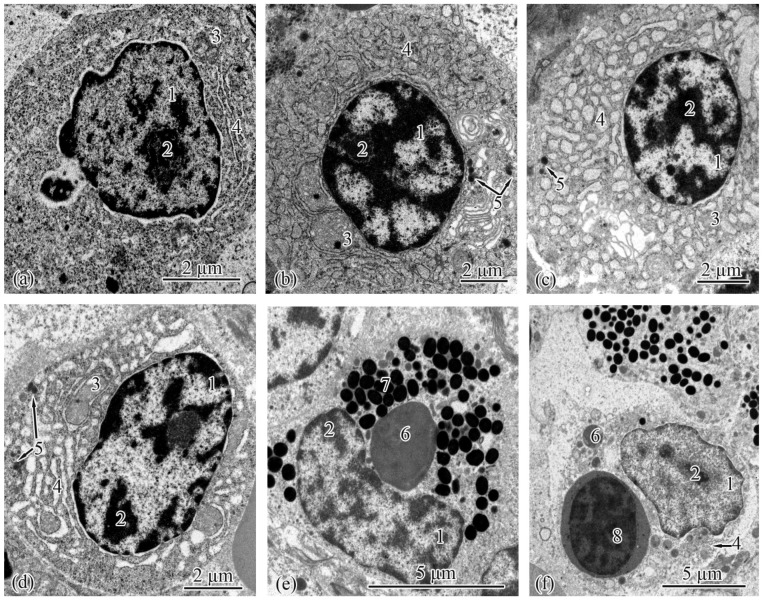
Ultrastructure of plasma cells and macrophages in the renal interstitia of the Atlantic salmon. Plasma cell in a parr from the Barents Sea population (**a**); plasma cell in a spawner from the Baltic Sea population (**b**); plasma cell in an adult M from the Barents Sea population (**c**); plasma cell in a spawner from the Barents Sea population (**d**); macrophage in a parr from the Baltic Sea population (**e**); macrophage in a spawner from the Baltic Sea population (**f**). Cell units: nucleus (1), heterochromatin (2), mitochondria (3), rough endoplasmic reticulum (4), lysosomes (5), phagosomes (6), melanin granules (7), cell in the cytoplasm of a macrophage (8).

**Figure 8 biology-12-00750-f008:**
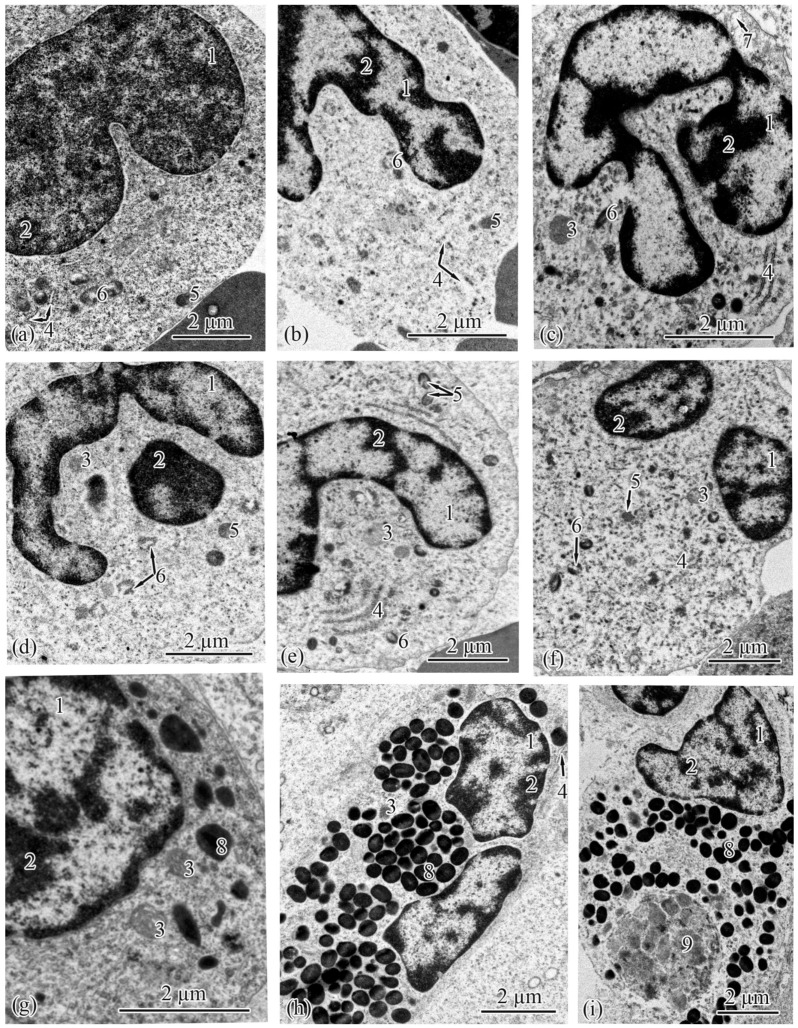
Ultrastructure of granulocytes in the renal interstitia of Atlantic salmon. Neutrophilic metamyelocyte in a parr from the Baltic Sea population (**a**); neutrophil with a horseshoe-shaped nucleus in a smolt from the Baltic Sea population (**b**); neutrophil with a horseshoe-shaped nucleus in an adult from the Baltic Sea population (**c**); neutrophil with a horseshoe-shaped nucleus in an adult S from the Barents Sea population (**d**); neutrophil with a horseshoe-shaped nucleus in an adult M from the Barents Sea population (**e**); neutrophil with a segmented nucleus in a spawner from the Baltic Sea population (**f**); eosinophil with a spherical nucleus in a parr from the Baltic Sea population (**g**); eosinophil with a segmented nucleus in a smolt from the Barents Sea population (**h**); eosinophil with a segmented nucleus in an adult M from the Barents Sea population (**i**). Cell units: nucleus (1), heterochromatin (2), mitochondria (3), rough endoplasmic reticulum (4), specific granules the first type of neutrophil (5), specific granules the second type of neutrophil (6), vesicle (7), specific granules of eosinophil (8), phagosome (9).

**Figure 9 biology-12-00750-f009:**
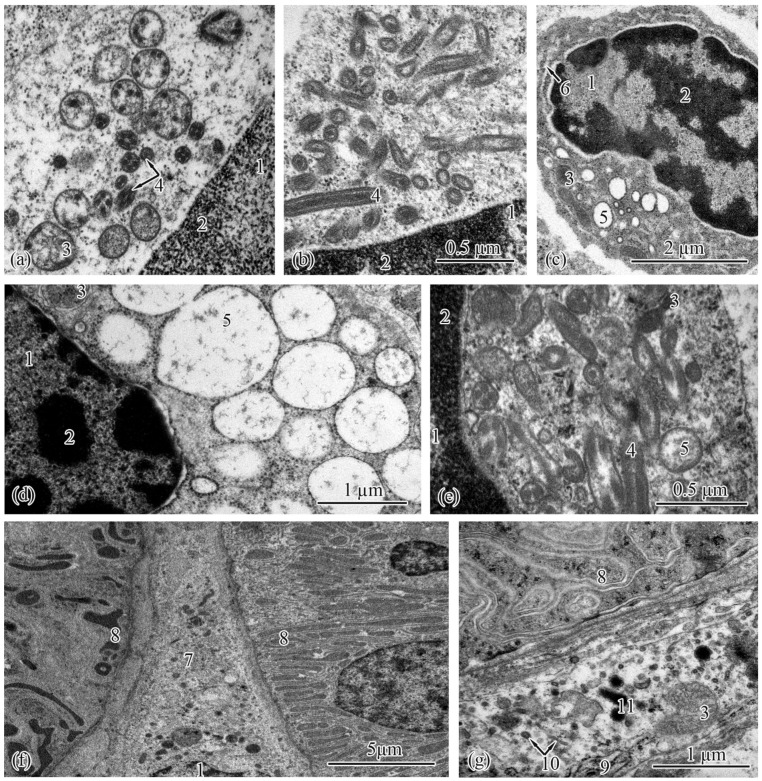
Ultrastructure of cells with radially arranged vesicles and chloride cells in the renal interstitia of Atlantic salmon. Cytoplasm of a cell with radially arranged vesicles in a parr from the Barents Sea population (**a**); cytoplasm of a cell with radially arranged vesicles in a smolt from the Baltic Sea population (**b**); cell with radially arranged vesicles in a smolt from the Baltic Sea population (**c**); cytoplasm of a cell with radially arranged vesicles in a spawner from the Baltic Sea population (**d**); cytoplasm of a cell with radially arranged vesicles in a spawner from the Barents Sea population (**e**); chloride cell in an adult S from the Barents Sea population (**f**); cytoplasm of a chloride cell in an adult S from the Barents Sea population (**g**). Cell units: nucleus (1), heterochromatin (2), mitochondria (3), radially arranged vesicles of fibrillar structure (4), electron-transparent radially arranged vesicles (5), rough endoplasmic reticulum (6), chloride cell (7), proximal tubules (8), microfibrils (9), tubular endoplasmic reticulum (10), electron-dense granules (11).

**Table 1 biology-12-00750-t001:** The renal interstitium proportion and outer diameters of the main structural units of trunk kidney in Atlantic salmon from the Baltic Sea population.

Ontogenetic Stage	*n*	Interstitium Proportion, %	Renal Corpuscle, μm	Proximal Tubule, μm	Distal Tubule, μm	Blood Vessel, μm
Parr	15	49.7 ± 3.08	81.7 ± 3.13 ^b^	60.7 ± 1.94 ^b^	58.5 ± 2.41 ^b^	20.5 ± 3.50
Smolts	12	51.2 ± 2.89	62.1 ± 4.47 ^a^	43.8 ± 1.22 ^a^	52.8 ± 2.51 ^a^	24.5 ± 2.45
Adults	6	48.2 ± 1.78	74.4 ± 3.32 ^a^	61.4 ± 2.99 ^b^	53.5 ± 1.95 ^a,b^	23.3 ± 2.83
Spawners	5	48.6 ± 1.55	82.1 ± 2.27 ^b^	61.6 ± 2.73 ^b^	57.3 ± 2.49 ^b^	23.9 ± 2.32

*n* = number of individuals. Data are given as mean ± SEM. Statistical differences between the studied ontogenetic stages are denoted by letter superscripts; means with different superscripts within columns have significant differences (Kruskal–Wallis ANOVA with post hoc Wilcoxon–Mann–Whitney test, *p* < 0.05).

**Table 2 biology-12-00750-t002:** The renal interstitium proportion and outer diameters of the main structural units of trunk kidney in Atlantic salmon from the Barents Sea population.

Ontogenetic Stage	*n*	Interstitium Proportion, %	Renal Corpuscle, μm	Proximal Tubule, μm	Distal Tubule, μm	Blood Vessel, μm
Parr	21	50.4 ± 2.25 ^a^	81.7 ± 3.13 ^b^	63.9 ± 1.55 ^b^	63.5 ± 1.67 ^c^	22.5 ± 1.70
Smolts	5	51.1 ± 2.23 ^a,b^	54.7 ± 2.69 ^a^	44.7 ± 1.14 ^a^	50.7 ± 1.71 ^b^	21.1 ± 1.40
Adults S	7	68.0 ± 1.45 ^c^	53.8 ± 2.69 ^a^	37.0 ± 1.07 ^a^	39.0 ± 0.98 ^a^	19.6 ± 2.02
Adults M	5	56.2 ± 2.16 ^b^	61.3 ± 2.92 ^a^	40.5 ± 1.79 ^a^	54.3 ± 2.82 ^b^	25.2 ± 5.75
Spawners	5	51.3 ± 1.83 ^a,b^	82.1 ± 2.62 ^b^	63.8 ± 1.45 ^b^	63.5 ± 1.51 ^c^	22.4 ± 0.93

*n* = number of individuals. Data are given as mean ± SEM. Statistical differences between the studied ontogenetic stages are denoted by letter superscripts; means with different superscripts within columns have significant differences (Kruskal–Wallis ANOVA with post hoc Wilcoxon–Mann–Whitney test, *p* < 0.05).

**Table 3 biology-12-00750-t003:** Morphometric parameters of the Bowman’s capsules in the trunk kidneys of Atlantic salmon from the Baltic Sea population.

Parameter	Parr (*n* = 15)	Smolts (*n* = 12)	Adults (*n* = 6)	Spawners (*n* = 5)
Podocyte area, μm^2^	46.0 ± 13.8	39.0 ± 4.00	48.0 ± 12.1	48.7 ± 4.44
Nucleus area, μm^2^	21.2 ± 13.5	11.2 ± 1.92	26.5 ± 7.01	27.2 ± 2.58
Urinary-space width, μm	5.69 ± 0.88 ^b^	1.15 ± 0.06 ^a^	1.04 ± 0.17 ^a^	3.29 ± 0.58 ^b^
Basement-membrane width, μm	0.13 ± 0.01 ^a^	0.43 ± 0.05 ^b^	0.40 ± 0.13 ^b^	0.18 ± 0.06 ^a^
Capillary diameter, μm	30.1 ± 4.43 ^b^	12.1 ± 1.61 ^a^	26.8 ± 2.03 ^b^	33.8 ± 0.40 ^b^

*n* = number of individuals. Data are given as mean ± SEM. Statistical differences between the studied ontogenetic stages are denoted by letter superscripts; means with different superscripts within rows have significant differences (Kruskal–Wallis ANOVA with post hoc Wilcoxon–Mann–Whitney test, *p* < 0.05).

**Table 4 biology-12-00750-t004:** Morphometric parameters of the Bowman’s capsules in the trunk kidneys of Atlantic salmon from the Barents Sea population.

Parameter	Parr (*n* = 21)	Smolts (*n* = 5)	Adults S (*n* = 7)	Adults M (*n* = 5)	Spawners (*n* = 5)
Podocyte area, μm^2^	44.7 ± 11.5	38.1 ± 2.36	29.7 ± 4.41	48.8 ± 10.8	48.7 ± 8.38
Nucleus area, μm^2^	16.7 ± 5.85	10.7 ± 1.12	17.8 ± 2.31	20.5 ± 2.81	19.7 ± 2.46
Urinary-space width, μm	5.22 ± 0.47 ^b^	1.58± 0.07 ^a^	1.00 ± 0.12 ^a^	3.84 ± 0.29 ^b^	4.38 ± 0.38 ^b^
Basement-membrane width, μm	0.18 ± 0.03 ^a^	0.38 ± 0.06 ^b^	0.52 ± 0.09 ^c^	0.44 ± 0.11 ^b,c^	0.27 ± 0.02 ^b^
Capillary diameter, μm	27.9 ± 3.02 ^b^	11.9 ± 0.61 ^a^	15.5 ± 0.66 ^a^	29.0 ± 0.74 ^b^	29.0 ± 0.82 ^b^

*n* = number of individuals. Data are given as mean ± SEM. Statistical differences between the studied ontogenetic stages are denoted by letter superscripts; means with different superscripts within rows have significant differences (Kruskal–Wallis ANOVA with post hoc Wilcoxon–Mann–Whitney test, *p* < 0.05).

**Table 5 biology-12-00750-t005:** Morphometric parameters of type I epithelial cells of the proximal tubule of the nephron in the trunk kidneys of Atlantic salmon from the Baltic Sea population.

Parameter	Parr (*n* = 15)	Smolts (*n* = 12)	Adults (*n* = 6)	Spawners (*n* = 5)
Cell area, μm^2^	94.4 ± 11.3 ^b^	60.3 ± 5.99 ^a^	67.6 ± 3.57 ^a^	116 ± 4.80 ^b^
Cell height, μm	17.4 ± 1.08 ^b^	14.0 ± 0.81 ^a^	14.8 ± 0.17 ^a^	18.8 ± 0.43 ^b^
Nucleus area, μm^2^	38.4 ± 6.38	27.1 ± 5.78	25.4 ± 8.23	31.5 ± 1.29
Mitochondrion area, μm^2^	0.30 ± 0.14 ^a^	0.77 ± 0.07 ^c^	0.76 ± 0.09 ^c^	0.58 ± 0.07 ^b^
Number of mitochondria; (min–max)	24.0 ± 0.35 ^a^ (22–26)	35.8 ± 2.20 ^b^ (31–43)	36.5 ± 2.13 ^b^ (31–41)	27.5 ± 1.36 ^a^ (19–39)
Secretory granule area, μm^2^	1.45 ± 0.04 ^b^	1.58 ± 0.27 ^b^	1.73 ± 0.35 ^b^	0.52 ± 0.02 ^a^
Number of secretory granules; (min–max)	3.17 ± 0.09 ^a^ (3–4)	9.50 ± 1.12 ^a^ (6–12)	22.7 ± 3.57 ^b^ (17–33)	19.9 ± 1.03 ^b^ (12–26)
Cisterna width of the smooth endoplasmic reticulum, μm	0.04 ± 0.00 ^a^	0.12 ± 0.01 ^b^	0.12 ± 0.02 ^b^	0.11 ± 0.02 ^b^
Endocytosis-zone length, μm	2.98 ± 0.12	3.38 ± 0.15	4.15 ± 0.21	3.97 ± 0.18
Brush-border length, μm	2.44 ± 0.08	2.41 ± 0.17	2.49 ± 0.16	2.75 ± 0.07
Microvillus diameter, μm	0.08 ± 0.00 ^a^	0.10 ± 0.01 ^b^	0.10 ± 0.00 ^b^	0.10 ± 0.00 ^b^
Cilium diameter, μm	0.21 ± 0.00	0.21 ± 0.00	0.21 ± 0.00	0.21 ± 0.00

*n* = number of individuals. Data are given as mean ± SEM. Statistical differences between the studied ontogenetic stages are denoted by letter superscripts; means with different superscripts within rows have significant differences (Kruskal–Wallis ANOVA with post hoc Wilcoxon–Mann–Whitney test, *p* < 0.05).

**Table 6 biology-12-00750-t006:** Morphometric parameters of type I epithelial cells of the proximal tubule of the nephron in the trunk kidneys of Atlantic salmon from the Barents Sea population.

Parameter	Parr (*n* = 21)	Smolts (*n* = 5)	Adults S (*n* = 7)	Adults M (*n* = 5)	Spawners (*n* = 5)
Cell area, μm^2^	105 ± 12.5 ^b^	60.3 ± 5.99 ^a^	66.9 ± 3.30 ^a^	74.5 ± 2.12 ^b^	73.0 ± 7.48 ^b^
Cell height, μm	18.5 ± 1.20 ^b^	14.9 ± 0.22 ^a^	14.7 ± 0.32 ^a^	15.2 ± 0.18 ^a^	18.3 ± 1.63 ^b^
Nucleus area, μm^2^	38.4 ± 7.50	26.6 ± 2.18	28.9 ± 7.39	31.1 ± 3.53	29.01 ± 4.09
Mitochondrion area, μm^2^	0.34 ± 0.14 ^a^	0.82 ± 0.05 ^b^	1.34 ± 0.13 ^c^	0.71 ± 0.05 ^b^	0.56 ± 0.07 ^a^
Number of mitochondria; (min–max)	25.5 ± 2.08 ^a^ (23–28)	36.2 ± 1.08 ^b^ (33–43)	39.4 ± 1.60 ^b^ (36–42)	35.0 ± 3.62 ^b^ (30–40)	27.5 ± 3.92 ^a^ (19–34)
Secretory granule area, μm^2^	1.47 ± 0.27 ^b^	1.61 ± 0.17 ^b^	1.90 ± 0.44 ^b^	0.44 ± 0.17 ^a^	0.50 ± 0.03 ^a^
Number of secretory granules; (min–max)	3.25 ± 0.50 ^a^ (3–4)	9.29 ± 0.59 ^a^ (6–12)	34.8 ± 1.24 ^c^ (32–38)	18.0 ± 1.74 ^b^ (11–22)	9.0 ± 1.41 ^a^ (6–12)
Cisterna width of the smooth endoplasmic reticulum, μm	0.04 ± 0.01 ^a^	0.12 ± 0.01 ^b^	0.15 ± 0.01 ^c^	0.11 ± 0.02 ^b^	0.11 ± 0.02 ^b^
Endocytosis-zone length, μm	3.09 ± 0.75	3.75 ± 0.18	4.11 ± 0.41	3.13 ± 0.38	2.25 ± 0.37
Brush-border length, μm	3.39 ± 0.45 ^b^	3.31 ± 0.17 ^b^	1.91 ± 0.33 ^a^	2.91 ± 0.17 ^b^	3.47 ± 0.30 ^b^
Microvillus diameter, μm	0.10 ± 0.01	0.10 ± 0.01	0.10 ± 0.01	0.11 ± 0.00	0.10 ± 0.00
Cilium diameter, μm	0.21 ± 0.02	0.21 ± 0.00	0.21 ± 0.00	0.21 ± 0.00	0.21 ± 0.00

*n* = number of individuals. Data are given as mean ± SEM. Statistical differences between the studied ontogenetic stages are denoted by letter superscripts; means with different superscripts within rows have significant differences (Kruskal–Wallis ANOVA with post hoc Wilcoxon–Mann–Whitney test, *p* < 0.05).

**Table 7 biology-12-00750-t007:** Morphometric parameters of type II epithelial cells of the proximal tubule of the nephron in the trunk kidneys of Atlantic salmon from the Baltic Sea population.

Parameter	Parr (*n* = 15)	Smolts (*n* = 12)	Adults (*n* = 6)	Spawners (*n* = 5)
Cell area, μm^2^	75.4 ± 1.79 ^a,b^	59.2 ± 5.68 ^a^	66.8 ± 5.16 ^a,b^	77.7 ± 3.91 ^b^
Cell height, μm	15.2 ± 0.15 ^a,b^	13.9 ± 0.71 ^a^	14.6 ± 0.42 ^a,b^	15.4 ± 0.12 ^b^
Nucleus area, μm^2^	29.3 ± 2.25	28.9 ± 8.10	25.2 ± 3.51	27.4 ± 2.12
Mitochondrion area, μm^2^	0.29 ± 0.03 ^a^	0.63 ± 0.06 ^b^	0.88 ± 0.06 ^c^	0.46 ± 0.06 ^a^
Number of mitochondria; (min–max)	25.0 ± 0.47 ^a^ (23–27)	49.0 ± 8.15 ^c^ (37–60)	31.6 ± 1.24 ^b^ (26–35)	21.8 ± 1.06 ^a^ (14–29)
Endocytosis-zone length, μm	1.35 ± 0.07	1.46 ± 0.09	1.52 ± 0.29	1.29 ± 0.02
Brush-border length, μm	2.68 ± 0.09	2.21 ± 0.33	2.30 ± 0.25	2.41 ± 0.15
Cilium diameter, μm	0.21± 0.00	0.21± 0.00	0.21± 0.02	0.21 ± 0.00
Microvillus diameter, μm	0.11 ± 0.01 ^a^	0.15 ± 0.00 ^b^	0.14 ± 0.02	0.14 ± 0.00

*n* = number of individuals. Data are given as mean ± SEM. Statistical differences between the studied ontogenetic stages are denoted by letter superscripts; means with different superscripts within rows have significant differences (Kruskal–Wallis ANOVA with post hoc Wilcoxon–Mann–Whitney test, *p* < 0.05).

**Table 8 biology-12-00750-t008:** Morphometric parameters of type II epithelial cells of the proximal tubule of the nephron in the trunk kidneys of Atlantic salmon from the Barents Sea population.

Parameter	Parr (*n* = 21)	Smolts (*n* = 5)	Adults S (*n* = 7)	Adults M (*n* = 5)	Spawners (*n* = 5)
Cell area, μm^2^	73.5 ± 7.03 ^b^	60.3 ± 6.32 ^a,b^	54.9 ± 2.38 ^a^	69.9 ± 2.87 ^a,b^	73.9 ± 8.63 ^b^
Cell height, μm	14.8 ± 0.34 ^b^	14.3 ± 0.26 ^b^	12.9 ± 0.26 ^a^	14.5 ± 0.27 ^b^	14.8 ± 0.41 ^b^
Nucleus area, μm^2^	30.0 ± 8.97	26.9 ± 1.92	24.3 ± 2.52	27.9 ± 1.59	26.8 ± 5.24
Mitochondrion area, μm^2^	0.30 ± 0.04 ^a^	0.54 ± 0.05 ^b^	1.34 ± 0.13 ^c^	1.28 ± 0.06 ^c^	0.50 ± 0.14 ^a,b^
Number of mitochondria; (min–max)	26.0 ± 1.03 ^a^ (23–28)	57.2 ± 8.15 ^b^ (50–67)	58.4 ± 2.14 ^b^ (49–68)	26.6 ± 1.01 ^a^ (25–32)	22.3 ± 2.21 ^a^ (19–29)
Endocytosis-zone length, μm	1.31 ± 0.05 ^a^	1.38 ± 0.14 ^a^	1.88 ± 0.21 ^b^	1.31 ± 0.07 ^a^	1.23 ± 0.05 ^a^
Brush-border length, μm	2.33 ± 0.46	2.21 ± 0.33	1.93 ± 0.09	2.31 ± 0.14	2.43 ± 0.14
Cilium diameter, μm	0.21± 0.01	0.21± 0.00	0.21± 0.00	0.21 ± 0.01	0.21 ± 0.00
Microvillus diameter, μm	0.12 ± 0.01	0.13 ± 0.00	0.13 ± 0.02	0.11 ± 0.01 ^a^	0.17 ± 0.02 ^b^

*n* = number of individuals. Data are given as mean ± SEM. Statistical differences between the studied ontogenetic stages are denoted by letter superscripts; means with different superscripts within rows have significant differences (Kruskal–Wallis ANOVA with post hoc Wilcoxon–Mann–Whitney test, *p* < 0.05).

**Table 9 biology-12-00750-t009:** Morphometric parameters of the distal-tubule epithelial cells of the nephron in the trunk kidneys of Atlantic salmon from the Baltic Sea population.

Parameter	Parr (*n* = 15)	Smolts (*n* = 12)	Adults (*n* = 6)	Spawners (*n* = 5)
Cell area, μm^2^	137 ± 2.85 ^b^	69.9 ± 6.35 ^a^	114 ± 14.6 ^b^	124 ± 8.10 ^b^
Cell height, μm	25.1 ± 0.30 ^b^	17.1 ± 0.55 ^a^	22.3 ± 0.13 ^b^	21.1 ± 0.34 ^b^
Nucleus area, μm^2^	44.1 ± 1.73 ^b^	18.9 ± 3.55 ^a^	21.4 ± 2.27 ^a^	20.9 ± 1.21 ^a^
Mitochondrion area, μm^2^	0.95 ± 0.07 ^b^	1.42 ± 0.30 ^c^	1.30 ± 0.25 ^b^	0.45 ± 0.03 ^a^
Number of mitochondria; (min–max)	31.5 ± 1.28 ^a^ (22–41)	62.2 ± 2.84 ^b^ (56–68)	42.2 ± 3.93 ^a^ (27–59)	33.3 ± 1.75 ^a^ (24–49)

*n* = number of individuals. Data are given as mean ± SEM. Statistical differences between the studied ontogenetic stages are denoted by letter superscripts; means with different superscripts within rows have significant differences (Kruskal–Wallis ANOVA with post hoc Wilcoxon–Mann–Whitney test, *p* < 0.05).

**Table 10 biology-12-00750-t010:** Morphometric parameters of the distal-tubule epithelial cells of the nephron in the trunk kidneys of Atlantic salmon from the Barents Sea population.

Parameter	Parr (*n* = 21)	Smolts (*n* = 5)	Adults S (*n* = 7)	Adults M (*n* = 5)	Spawners (*n* = 5)
Cell area, μm^2^	126 ± 4.88 ^b^	68.9 ± 2.90 ^a^	82.2 ± 8.44 ^a^	115 ± 7.95 ^b^	125 ± 2.45 ^b^
Cell height, μm	24.3 ± 0.36 ^c^	16.6 ± 0.15 ^a^	16.5 ± 0.54 ^a,b^	22.0 ± 0.94 ^b^	21.1 ± 0.74 ^b^
Nucleus area, μm^2^	44.3 ± 5.21 ^b^	18.0 ± 1.44 ^a^	17.0 ± 2.59 ^a^	28.9 ± 2.65 ^a^	26.9 ± 4.62 ^a^
Mitochondrion area, μm^2^	0.96 ± 0.11 ^b^	1.57 ± 0.20 ^c^	1.70 ± 0.20 ^c^	0.90 ± 0.33 ^b^	0.44 ± 0.07 ^a^
Number of mitochondria; (min–max)	32.1 ± 2.18 ^a^ (22–41)	62.3 ± 0.80 ^b^ (58–68)	63.0 ± 1.45 ^b^ (59–71)	33.2 ± 4.60 ^a^ (29–56)	34.1 ± 4.32 ^a^ (24–49)

*n* = number of individuals. Data are given as mean ± SEM. Statistical differences between the studied ontogenetic stages are denoted by letter superscripts; means with different superscripts within rows have significant differences (Kruskal–Wallis ANOVA with post hoc Wilcoxon–Mann–Whitney test, *p* < 0.05).

**Table 11 biology-12-00750-t011:** Morphometric parameters of agranulocytes in the trunk kidneys of Atlantic salmon from the Baltic Sea population.

Leukocyte Type	Ontogenetic Stage	*n*	Cell Area, μm^2^	Nucleus Area, μm^2^	Mitochondrion Area, μm^2^	Number of Mitochondria (Min–Max)
Lymphocyte	Parr	15	18.5 ± 4.22 ^a^	15.8 ± 2.49	0.14 ± 0.08 ^a^	1.40 ± 0.32 ^a^; (1–2)
	Smolts	12	27.1 ± 3.86 ^a,b^	16.0 ± 0.68	0.26 ± 0.04 ^a,b^	4.20 ± 0.75 ^b^; (2–5)
	Adults	6	32.5 ± 3.34 ^b^	16.3 ± 2.00	0.39 ± 0.10 ^b^	3.80 ± 0.26 ^b^; (3–4)
	Spawners	5	17.9 ± 1.07 ^a^	10.4 ± 0.67	0.16 ± 0.02 ^a^	2.50 ± 0.24 ^a,b^; (2–3)
Plasma cell	Parr	15	62.8 ± 5.63 ^a^	30.2 ± 4.19	0.26 ± 0.03 ^a^	1.17 ± 0.24 ^a^; (1–2)
	Smolts	12	73.5 ± 5.20 ^a,b^	20.3 ± 3.59	0.36 ± 0.04 ^a^	6.80 ± 0.63 ^b^; (5–7)
	Adults	6	85.5 ± 3.43 ^b^	24.5 ± 1.94	1.10 ± 0.12 ^b^	6.00 ± 0.28 ^b^; (5–7)
	Spawners	5	59.0 ± 6.00 ^a^	25.2 ± 3.12	0.17 ± 0.03 ^a^	1.38 ± 0.23 ^a^; (1–2)
Macrophage	Parr	15	113 ± 8.69 ^a,b^	22.2 ± 5.72 ^b^	0.36 ± 0.12 ^a^	1.33 ± 0.30 ^a^; (1–2)
	Smolts	12	101 ± 13.8 ^a,b^	16.0 ± 2.05 ^a,b^	1.02 ± 0.12 ^b^	6.25 ± 0.55 ^b^; (5–7)
	Adults	6	130 ± 7.60 ^b^	9.39 ± 0.58 ^a^	0.85 ± 0.14 ^b^	5.17 ± 0.58 ^b^; (4–7)
	Spawners	5	92.1 ± 8.81 ^a^	14.0 ± 4.56 ^a,b^	0.27 ± 0.09 ^a^	2.29 ± 0.44 ^a^; (1–3)

*n* = number of individuals. Data are given as mean ± SEM. For each leukocyte type, statistical differences between the studied ontogenetic stages are denoted by letter superscripts; means with different superscripts within columns have significant differences (Kruskal–Wallis ANOVA with post hoc Wilcoxon–Mann–Whitney test, *p* < 0.05).

**Table 12 biology-12-00750-t012:** Morphometric parameters of agranulocytes in the trunk kidneys of Atlantic salmon from the Barents Sea population.

Leukocyte Type	Ontogenetic Stage	*n*	Cell Area, μm^2^	Nucleus Area, μm^2^	Mitochondrion Area, μm^2^	Number of Mitochondria (Min–Max)
Lymphocyte	Parr	21	19.1 ± 5.15 ^a^	14.0 ± 3.81	0.18 ± 0.08 ^a^	2.00 ± 0.58 ^a^; (1–3)
	Smolts	5	29.2 ± 2.16 ^a^	15.3 ± 1.06	0.25 ± 0.05 ^a,b^	4.29 ± 0.80 ^b^; (2–6)
	Adults S	7	32.4 ± 2.85 ^b^	13.1 ± 1.80	0.49 ± 0.11 ^b^	4.00 ± 0.58 ^b^; (3–5)
	Adults M	5	22.7 ± 4.54 ^a^	12.2 ± 2.99	0.30 ± 0.05 ^b^	2.75 ± 0.35 ^a,b^; (2–3)
	Spawners	5	18.6 ± 1.03 ^a^	10.4 ± 0.78	0.15 ± 0.02 ^a^	2.83 ± 0.34 ^a,b^; (2–4)
Plasma cell	Parr	21	65.5 ± 7.32	27.2 ± 5.76	0.26 ± 0.04 ^a^	1.25 ± 0.29 ^a^; (1–2)
	Smolts	5	72.7 ± 4.64	20.0 ± 3.12	0.34 ± 0.04 ^a^	7.40 ± 1.05 ^b^; (5–9)
	Adults S	7	87.4 ± 5.18	23.4 ± 1.01	1.03 ± 0.25 ^b^	6.00 ± 0.87 ^b^; (5–8)
	Adults M	5	80.7 ± 1.77	22.9 ± 4.37	0.64 ± 0.17 ^a,b^	4.80 ± 0.17 ^a,b^; (2–6)
	Spawners	5	60.6 ± 6.92	26.8 ± 3.48	0.18 ± 0.04 ^a^	1.33 ± 0.23 ^a,b^; (1–2)
Macrophage	Parr	21	107 ± 9.05 ^a,b^	24.5 ± 9.10	0.33 ± 0.15 ^a^	1.33 ± 0.33 ^a^; (1–2)
	Smolts	5	115 ± 13.0 ^b^	15.5 ± 1.87	1.01 ± 0.10 ^b^	7.17 ± 0.77 ^b^; (5–9)
	Adults S	7	132 ± 22.7 ^b^	11.8 ± 6.77	0.92 ± 0.14 ^b^	5.40 ± 0.63 ^b^; (4–6)
	Adults M	5	117 ± 11.8 ^b^	24.5 ± 11.1	0.73 ± 0.17 ^b^	4.40 ± 0.81 ^b^; (3–6)
	Spawners	5	77.2 ± 10.4 ^a^	15.3 ± 5.25	0.27 ± 0.10 ^a^	2.40 ± 0.51 ^a^; (1–4)

*n* = number of individuals. Data are given as mean ± SEM. For each leukocyte type, statistical differences between the studied ontogenetic stages are denoted by letter superscripts; means with different superscripts within columns have significant differences (Kruskal–Wallis ANOVA with post hoc Wilcoxon–Mann–Whitney test, *p* < 0.05).

**Table 13 biology-12-00750-t013:** Morphometric parameters of granulocytes in the trunk kidneys of Atlantic salmon from the Baltic Sea population.

Leukocyte Type	Ontogenetic Stage	*n*	Cell Area, μm^2^	Nucleus Area, μm^2^	Mitochondrion Area, μm^2^	Number of Mitochondria (Min–Max)	Granule Area, μm^2^	Number of Granules (Min–Max)
Neutrophil	Parr	15	53.9 ± 1.40 ^a^	11.1 ± 3.78	0.07± 0.01 ^a^	3.00 ± 0.24 ^a^ (2–4)	0.14 ± 0.02	4.14 ± 0.70 ^a^ (3–6)
	Smolts	12	57.1 ± 6.62 ^a^	13.6 ± 2.18	0.20 ± 0.03 ^b^	6.00 ± 0.41 ^b^ (5–7)	0.08 ± 0.03	6.50 ± 0.71 ^b^ (5–8)
	Adults	6	62.7 ± 8.65 ^a,b^	13.0 ± 0.81	3.82 ± 1.47 ^c^	5.43 ± 0.57 ^b^ (4–7)	0.10 ± 0.02	6.71 ± 0.31 ^b^ (6–8)
	Spawners	5	76.1 ± 5.19 ^b^	15.0 ± 4.72	5.13 ± 1.61 ^c^	5.60 ± 0.41 ^b^ (4–7)	0.14 ± 0.08	10.4 ±1.38 ^c^ (8–13)
Eosinophil	Parr	15	64.9 ± 10.5 ^a^	15.6 ± 1.93^a^	0.11 ± 0.01 ^a^	1.00 ± 0.00 ^a^ (1)	0.22 ± 0.02 ^b^	29.6 ± 2.76 ^a^ (22–35)
	Smolts	12	86.7 ± 10.7 ^b^	14.2 ± 1.50^a^	0.69 ± 0.06 ^b^	5.20 ±0.75 ^b^ (4–7)	0.18 ± 0.03 ^b^	86.0 ± 10.5 ^b^ (64–108)
	Adults	6	94.0 ± 7.63 ^b^	12.2 ± 2.06^a^	1.08 ± 0.11 ^b^	6.00 ± 1.18 ^b^ (4–8)	0.20 ± 0.03 ^b^	73.7 ± 8.09 ^b^ (60–101)
	Spawners	5	93.8 ± 6.25 ^b^	8.18 ± 1.03^b^	0.78 ± 0.19 ^b^	2.67 ± 0.47 ^a^ (2–4)	0.15 ± 0.02 ^a^	85.2 ± 12.1 ^b^ (64–102)

*n* = number of individuals. Data are given as mean ± SEM. For each leukocyte type, statistical differences among the studied ontogenetic stages are denoted by letter superscripts; means with different superscripts within columns have significant differences (Kruskal–Wallis ANOVA with post hoc Wilcoxon–Mann–Whitney test, *p* < 0.05).

**Table 14 biology-12-00750-t014:** Morphometric parameters of granulocytes in the trunk kidney of Atlantic salmon from the Barents Sea population.

Leukocyte Type	Ontogenetic Stage	*n*	Cell Area, μm^2^	Nucleus Area, μm^2^	Mitochondrion Area, μm^2^	Number of Mitochondria (Min–Max)	Granule Area, μm^2^	Number of Granules (Min–Max)
Neutrophil	Parr	21	53.1 ± 1.75 ^a^	12.4 ± 4.63	0.08 ± 0.01 ^a^	3.33 ± 0.33 ^a^ (3–4)	0.15 ± 0.02	4.20 ± 0.75 ^a^ (3–6)
	Smolts	5	67.4 ± 6.86 ^b^	13.3 ± 1.73	0.22 ± 0.04 ^b^	5.71 ± 0.55 ^b^ (4–7)	0.08 ± 0.03	6.80 ± 0.71 ^b^ (5–9)
	Adults S	7	76.1 ± 5.19 ^b^	15.0 ± 4.72	5.13 ± 1.61 ^c^	5.60 ± 0.95 ^b^ (4–7)	0.14 ± 0.08	10.4 ±1.38 ^b^ (8–13)
	Adults M	5	48.8 ± 2.38 ^a,b^	10.9 ± 1.62	0.17 ± 0.03 ^b^	4.80 ± 0.77 ^b^ (4–6)	0.13 ± 0.02	10.4 ± 2.19 ^b^ (7–16)
	Spawners	5	47.5 ± 1.27 ^a^	11.0 ± 1.83	0.09 ± 0.01 ^a^	3.60 ± 0.24 ^a^ (3–4)	0.14 ± 0.01	11.8 ± 1.53 ^b^ (7–15)
Eosinophil	Parr	21	63.5 ± 13.4 ^a^	16.9 ± 2.89 ^a^	0.10 ± 0.01 ^a^	1.00 ± 0.00 ^a^ (1)	0.24 ± 0.03 ^a^	16.0 ± 9.85 ^a^ (12–35)
	Smolts	5	86.7 ± 11.4 ^b^	13.5 ± 1.68	0.67 ± 0.05 ^b^	2.75 ± 0.51 ^b^ (2–4)	0.18 ± 0.02 ^a^	83.8 ± 8.67 ^b^ (64–108)
	Adults S	7	93.8 ± 6.51 ^b^	8.18 ± 1.03 ^b^	1.08 ± 0.11 ^c^	6.00 ± 1.18 ^c^ (4–8)	0.15 ± 0.02 ^a^	85.2 ± 12.1 ^b^ (64–102)
	Adults M	5	74.3 ± 12.8 ^a^	12.5 ± 2.77	0.17 ± 0.03 ^a^	4.20 ± 0.59 ^b^ (3–5)	0.20 ± 0.03 ^a^	88.7 ± 8.41 ^b^ (71–103)
	Spawners	5	55.6 ± 6.78 ^a^	13.1 ± 2.17	0.09 ± 0.01 ^a^	2.67 ± 0.23 ^b^ (2–3)	0.83 ± 0.05 ^b^	96.0 ± 4.25 ^b^ (84–112)

*n* = number of individuals. Data are given as mean ± SEM. For each leukocyte type, statistical differences between the studied ontogenetic stages are denoted by letter superscripts; means with different superscripts within columns have significant differences (Kruskal–Wallis ANOVA with post hoc Wilcoxon–Mann–Whitney test, *p* < 0.05).

**Table 15 biology-12-00750-t015:** Morphometric parameters of cells with radially arranged vesicles and chloride cells in the trunk kidneys of Atlantic salmon from the Baltic Sea population.

Cell Type	Ontogenetic Stage	*n*	Cell Area, μm^2^	Nucleus Area, μm^2^	Mitochondrion Area, μm^2^	Number of Mitochondria (Min–Max)	Vesicle Area, μm^2^	Number of Granules (Min–Max)
Cell with radially arranged vesicles	Parr	15	n.f.	n.f.	n.f.	n.f.	n.f.	n.f.
	Smolts	12	28.8 ± 3.71	10.4 ± 0.79	0.49 ± 0.05 ^a^	6.40 ± 0.52 ^a^ (5–7)	0.03 ± 0.01 ^a^	6.40 ± 0.52 (5–7)
	Adults	6	n.f.	n.f.	n.f.	n.f.	n.f.	n.f.
	Spawners	5	20.4 ± 1.82	11.1 ± 1.71	0.19 ± 0.01 ^b^	3.20 ± 0.22 ^b^ (3–4)	0.30 ± 0.04 ^b^	5.75 ± 0.73 (4–7)
Chloride cell	Parr	15	n.f.	n.f.	n.f.	n.f.	n.f.	n.f.
	Smolts	12	61.3 ± 10.2 ^a^	14.2 ± 1.50 ^a^	0.26 ± 0.04 ^b^	20.9 ± 1.23 ^b^ (17–24)	n.a.	n.a.
	Adults	6	120 ± 0.00 ^b^	22.3 ± 1.16 ^b^	1.25 ± 0.34 ^c^	38.7 ±1.45 ^c^ (36–41)	n.a.	n.a.
	Spawners	5	55.4 ± 2.80 ^a^	12.1 ± 0.30 ^a^	0.13 ± 0.02 ^a^	7.25 ± 0.47 ^a^ (5–9)	n.a.	n.a.

*n* = number of individuals; n.f. = not found, n.a. = not applicable. Data are given as mean ± SEM. For each cell type, statistical differences between the studied ontogenetic stages are denoted by letter superscripts; means with different superscripts within columns have significant differences (Kruskal–Wallis ANOVA with post hoc Wilcoxon–Mann–Whitney test, *p* < 0.05).

**Table 16 biology-12-00750-t016:** Morphometric parameters of cells with radially arranged vesicles and chloride cells in the trunk kidneys of Atlantic salmon from the Barents Sea population.

Cell Type	Ontogenetic Stage	*n*	Cell Area, μm^2^	Nucleus Area, μm^2^	Mitochondrion Area, μm^2^	Number of Mitochondria (Min–Max)	Vesicle Area, μm^2^	Number of Granules (Min–Max)
Cell with radially arranged vesicles	Parr	21	26.6 ± 4.23	13.6 ± 2.26	0.14 ± 0.00 ^a^	2.80 ± 0.48 ^a^ (2–4)	0.03 ± 0.01 ^a^	6.00 ± 1.56 ^a^ (4–10)
	Smolts	5	28.8 ± 3.32	10.4 ± 0.65	0.52 ± 0.06 ^b^	6.17 ± 0.57 ^b^ (5–7)	0.23 ± 0.09 ^b^	39.6 ± 0.99 ^b^ (38–42)
	Adults S	7	n.f.	n.f.	n.f.	n.f.	n.f.	n.f.
	Adults M	5	n.f.	n.f.	n.f.	n.f.	n.f.	n.f.
	Spawners	5	20.4 ± 1.82	11.1 ± 1.71	0.19 ± 0.01 ^a^	3.20 ± 0.22 ^a^ (3–4)	0.30 ± 0.04 ^b^	5.75 ± 0.73 ^a^ (4–7)
Chloride cell	Parr	21	n.f.	n.f.	n.f.	n.f.	n.f.	n.f.
	Smolts	5	62.0 ± 1.66 ^a^	14.3 ± 1.72 ^a^	0.25 ± 0.04 ^b^	22.0 ± 1.85 ^b^ (17–26)	n.a.	n.a.
	Adults S	7	125 ± 3.41 ^b^	23.0 ± 0.49 ^b^	1.07 ± 0.53 ^c^	40.7 ± 1.78 ^c^ (38–43)	n.a.	n.a.
	Adults M	5	57.7 ± 8.12 ^a^	13.2 ± 2.11 ^a^	0.29 ± 0.14 ^b^	14.7 ± 2.67 ^b^ (9–18)	n.a.	n.a.
	Spawners	5	53.7 ± 5.26 ^a^	13.2 ± 1.44 ^a^	0.16 ± 0.05 ^a^	8.17 ± 1.32 ^a^ (6–13)	n.a.	n.a.

*n* = number of individuals; n.f. = not found, n.a. = not applicable. Data are given as mean ± SEM. For each cell type, statistical differences between the studied ontogenetic stages are denoted by letter superscripts; means with different superscripts within columns have significant differences (Kruskal–Wallis ANOVA with post hoc Wilcoxon–Mann–Whitney test, *p* < 0.05).

## Data Availability

The datasets used in this study are publicly available at: https://disk.yandex.ru/d/YpkhRfDMSR9ArQ, accessed on 20 April 2023.

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
