# Peer review of "Histology and Ultrastructure of the Nephron and Kidney Interstitial Cells in the Atlantic Salmon (Salmo salar Linnaeus 1758) at Different Stages of Life Cycle"

_biology, 2023, doi:10.3390/biology12050750_

Round 1

Reviewer 1 Report

Dear,

       This manuscript is good to write; however, some minors are found. Could you please see the attached fie?

Best regards,

Could you please see the attached file?

Best regards,

Author Response

We would like to express our deepest gratitude to the experts for their valuable suggestions and efforts to improve the manuscript. The list of our responses is given below.

Comment: it is hardly understand pls use kidney or mesonephros? [Title]

Reply: The term “mesonephros”, a synonym for “trunk kidney”, is in common usage in the literature on fish biology. Therefore, we feel confident to use both these terms throughout the text. However, since the term “kidney” is more popular for running a query on a search engine, we used it in a title.

Comment: Italic = scientific name

Reply: Italics was checked throughout the text; all binomial names were italicized.

Comment: The title is too long and I suggested to delete this phrase.

Reply: The title was revised accordingly.

Comment: what is the new data from this research? [Simple Summary section]

Reply: Simple Summary section was revised.

Comment: add the country? [Baltic and Barents Seas]

Reply: The Baltic Sea is an arm of the Atlantic Ocean that is enclosed by Denmark, Estonia, Finland, Germany, Latvia, Lithuania, Poland, Russia, Sweden and the Barents Sea is a marginal sea of the Arctic Ocean, located off the northern coasts of Norway and Russia and divided between Norwegian and Russian territorial waters; as in the sea, salmons travel between feeding grounds, we do not believe that the country must be mentioned here.

Comment: pls add references [First paragraph of the Introduction].

Reply: Relevant references were added (1-3).

Comment: This paragraph should be added the benefits from fish kidney, and add the previous data on the salmon life cycle? [Second paragraph in the Introduction section].

Reply: The paragraph was changed.

Comment: why you are focus on this area? [the Baltic Sea and Barents Sea basins].

Reply: Relevant text was added in L131.

Comment: More details for this figure is required in association with the sub-figures [Figure 1].

Reply: Figure caption was revised.

Reviewer 2 Report

Entitle manuscript “Microanatomy and Ultrastructure of the nephron and kidney interstitial cells in the Atlantic salmon (Salmo salar Linnaeus 1758) at different stages of life cycle from the population of the Baltic and Barents Seas” by Ekaterina A. Flerova et al., provide in depth data by analyzing TEM results about kidney interstitial cells. No doubt the data presented here are highly significant but seems the length of the manuscript is too long which highly recommended to precisely presented. Authors are suggested to only highlight their significant results and discuss them. Presenting all the cellular data lengthens the manuscript. Also review the language of manuscript for the clarity of manuscript. At current manuscript needed moderate revision to be selected in future.

Some comments are suggested below,

·         Make the bold version of fonts “Table” and “Figure” throughout the manuscript.

·         Provide abbreviation of “ARRIVE.”

·         Give details of chemicals/equipment’s (Batch; Company; City; Country) used in current manuscript.

·         What was the temperature of post fixation in osmium tetroxide? And accelerating voltage of TEM?

·         Define “adult S” and “adult M” somewhere at the starting of manuscript where it was first used.

·         It will be good if the author provides the google location with aptitude of study area and try to concisely define the study area.

·         Better to give separate subheading of paragraph from line 140 to 148.

·         In all figure, assigned number should be colored with noted theme so it can be found easily. However, in some figures it makes hard to identify the numbers.

·         Conclusion section here in manuscript is presented like summary. Make it short with main findings and conclude it in about maximum 4-5 lines.

Already mentioned in above section

Author Response

We would like to express our deepest gratitude to the experts for their valuable suggestions and efforts to improve the manuscript. The list of our responses is given below.

Comment: No doubt the data presented here are highly significant but seems the length of the manuscript is too long which highly recommended to precisely presented. Authors are suggested to only highlight their significant results and discuss them. Presenting all the cellular data lengthens the manuscript.

Reply: We value this opinion; however, three other reviewers did not suggest any reductions to the manuscript, and we could not decide which data to exclude. Therefore, we respectfully leave the decision to Editor whether to reduce the Result section or not.

Comment: Also review the language of manuscript for the clarity of manuscript.

Reply: The manuscript has been additionally checked by a colleague fluent in English writing.

Comment: Make the bold version of fonts “Table” and “Figure” throughout the manuscript.

Reply: The revisions were made accordingly.

Comment: Provide abbreviation of “ARRIVE.”

Reply: The revision was made.

Comment: Give details of chemicals/equipment’s (Batch; Company; City; Country) used in current manuscript.

Reply: The revisions were made accordingly.

Comment: What was the temperature of post fixation in osmium tetroxide? And accelerating voltage of TEM?

Reply: The temperature of post fixation in osmium tetroxide was 20 ± 2 °C; accelerating voltage of TEM was 80 kV. These data were added (LL 217 and 239).

Comment: Define “adult S” and “adult M” somewhere at the starting of manuscript where it was first used.

Reply: These notations are defined in the M&M section (LL 195-203).

Comment: It will be good if the author provides the google location with aptitude of study area and try to concisely define the study area

Reply: Altitudes of the sampling sites were added (LL 192-202). Study areas are described in the Study area subsection.

Comment: Better to give separate subheading of paragraph from line 140 to 148.

Reply: The section is titled “Fish and Sampling” (a commonly used heading); this paragraph describes numbers of fish specimens used and is followed by a short paragraph on kidney samples preparation; therefore, we do not believe that a separate subheading is needed for the paragraph in question.

Comment: Conclusion section here in manuscript is presented like summary. Make it short with main findings and conclude it in about maximum 4-5 lines.

Reply: Conclusion section was reduced.

Reviewer 3 Report

I am writing about the manuscript entitle: Microanatomy and ultrastructure of the nephron and kidney interstitial cells in the Atlantic salmon (Salmo salar Linnaeus 1758) at different stages of life cycle from the populations of the Baltic and Barents Seas. In the present study the authors described the microanatomy and ultrastructure of Atlantic salmon from the Baltic and Barents Sea populations comparing some ontogenetic stages: parr, smolts, adults living at sea, adults returning to a natal river to spawn, and spawners. The study shows ultrastructural changes in the renal corpuscle and cells of the proximal tubules; furthermore changes were observed in the immune cells that occupy the renal interstitium. These changes may be related to ion transport, hyperosmotic conditions, energy production.

The paper is very interesting, and the study is very detailed. The manuscript is well-designed and well-written. Materials, Methods and Figures are clear and organized. In my opinion the ms can be accepted for publication after a few corrections regarding the italics of the genus and species considered and the conclusions that are too long.

Title

Please, write in italics Salmo salar

Introduction

Lines: 66 and 69 write in italics salmo trutta 

Lines:  72-73 and 79 write in italics salmo salar 

Results

Line 684: correct Lepeophtheirus salmonis 

Line: 690 write in italics Salmo salar 

Line: 712 write in italics Oncorhynchus mykiss.

Author Response

We would like to express our deepest gratitude to the experts for their valuable suggestions and efforts to improve the manuscript. The list of our responses is given below.

Comment: In my opinion the ms can be accepted for publication after a few corrections regarding the italics of the genus and species considered and the conclusions that are too long.

Reply: Italics was checked throughout the text; revisions were made accordingly, and the Conclusion section was reduced.

Reviewer 4 Report

Firstly, I want to note the high quality of the work done, the large number of measurements made, and the excellent illustrations. In general, I really liked this work, but I have a few small comments.

I believe that histological illustrations would be very suitable in this work, especially since there are histological descriptions of cells (especially their form and position, which readers can’t see at ultra magnification) in the text.

The term “Microanatomy” is not entirely accurate for describing stained cell sections. The term "histology" is not only in this case a synonym for the term "microanatomy", but also better describes the methods and results of the study. Also, the term "histology" is more popular, and it will be more convenient for readers to find this article on the Internet.

Something strange with Latin names of fish and invertebrate species. In the introduction and results, they are in normal type, in the discussion in italics. In line 684 – in both types. I offer to fix it.

Introduction. Lines 277-291. The aim of the work is confusing since this sentence is followed by information about alleles and colonization. It is better and more convenient for readers when the aim is at the end of the paragraph.

Tables. “Different letter superscripts within rows indicate statistically significant differences”

Couldn't find which one statistically significant indicate a and b. a - means there is statistically significant or not? I offered to add this in the table description because it is unclear.

Author Response

We would like to express our deepest gratitude to the experts for their valuable suggestions and efforts to improve the manuscript. The list of our responses is given below.

Comment: I believe that histological illustrations would be very suitable in this work, especially since there are histological descriptions of cells (especially their form and position, which readers can’t see at ultra magnification) in the text.

Reply: Histological illustrations were added (please, see Figure 2).

Comment: The term “Microanatomy” is not entirely accurate for describing stained cell sections. The term "histology" is not only in this case a synonym for the term "microanatomy", but also better describes the methods and results of the study. Also, the term "histology" is more popular, and it will be more convenient for readers to find this article on the Internet.

Reply: The revisions have been made accordingly.

Comment: Something strange with Latin names of fish and invertebrate species. In the introduction and results, they are in normal type, in the discussion in italics. In line 684 – in both types. I offer to fix it.

Reply: Italics was checked throughout the text.

Comment: Introduction. Lines 277-291. The aim of the work is confusing since this sentence is followed by information about alleles and colonization. It is better and more convenient for readers when the aim is at the end of the paragraph.

Reply: The paragraph was rearranged accordingly.

Comment: Tables. “Different letter superscripts within rows indicate statistically significant differences”. Couldn't find which one statistically significant indicate a and b. a - means there is statistically significant or not? I offered to add this in the table description because it is unclear.

Reply: Letter superscripts denote statistically homogenous groups; means with the same superscripts do not have significant differences, e.g., mean1±SEMa and mean2±SEMa do not differ between each other, while they differ from mean3±SEMb, whereas mean4±SEMa,b does not differ from either mean1, mean2 or mean3. Table descriptions was revised.